# β-lactamase expression induces collateral sensitivity in *Escherichia coli*

Cristina Herencias[1,2,8] ✉, Laura Álvaro-Llorente [1,8], Paula Ramiro-Martínez[1], Ariadna Fernández-Calvet [3], Ada Muñoz-Cazalla[1], Javier DelaFuente [3], Fabrice E. Graf [4,5,6], Laura Jaraba-Soto[1], Juan Antonio Castillo-Polo [1], Rafael Cantón [1,2], Álvaro San Millán [3,7] ✉ & Jerónimo Rodríguez-Beltrán [1,2] ✉

Major antibiotic groups are losing effectiveness due to the uncontrollable spread of antimicrobial resistance (AMR) genes. Among these, β-lactam resistance genes –encoding β-lactamases– stand as the most common resistance mechanism in Enterobacterales due to their frequent association with mobile genetic elements. In this context, novel approaches that counter mobile AMR are urgently needed. Collateral sensitivity (CS) occurs when the acquisition of resistance to one antibiotic increases susceptibility to another antibiotic and can be exploited to eliminate AMR selectively. However, most CS networks described so far emerge as a consequence of chromosomal mutations and cannot be leveraged to tackle mobile AMR. Here, we dissect the CS response elicited by the acquisition of a prevalent antibiotic resistance plasmid to reveal that the expression of the β-lactamase gene $bla_{OXA-48}$ induces CS to colistin and azithromycin. We next show that other clinically relevant mobile β-lactamases produce similar CS responses in multiple, phylogenetically unrelated *E. coli* strains. Finally, by combining experiments with surveillance data comprising thousands of antibiotic susceptibility tests, we show that β-lactamase-induced CS is pervasive within Enterobacterales. These results highlight that the physiological side-effects of β-lactamases can be leveraged therapeutically, paving the way for the rational design of specific therapies to block mobile AMR or at least counteract their effects.

The continuous evolution of antimicrobial resistance (AMR) is outpacing human efforts to control bacterial infections. Antibiotic-resistant bacteria are a severe threat to public health, leading to more than one million deaths yearly[1]. Due to the lack of new antibiotics in the development pipeline, new strategies are urgently needed to block or reduce the dissemination of AMR. One promising strategy is the exploitation of collateral sensitivity (CS), a phenomenon by which

the acquisition of resistance to one antibiotic renders bacteria more sensitive to a second antibiotic[2,3]. Rationally designed treatments that combine antibiotics with reciprocal CS promise to eradicate bacterial infections and constrain the evolution of AMR[3,4].

Multiple studies have uncovered CS networks associated with mutations in chromosomal genes across various bacterial species. For example, certain mutations conferring aminoglycoside resistance

[1]Servicio de Microbiología, Instituto Ramón y Cajal de Investigación Sanitaria (IRYCIS), Hospital Universitario Ramón y Cajal, Madrid, Spain. [2]Centro de Investigación Biomédica en Red de Enfermedades Infecciosas-CIBERINFEC, Instituto de Salud Carlos III, Madrid, Spain. [3]Centro Nacional de Biotecnología-CSIC, Madrid, Spain. [4]Department of Chemistry and Molecular Biology, University of Gothenburg, Gothenburg, Sweden. [5]Centre for Antibiotic Resistance Research (CARe), University of Gothenburg, Gothenburg, Sweden. [6]Department of Clinical Sciences, Liverpool School of Tropical Medicine, Liverpool, UK. [7]Centro de Investigación Biológica en Red de Epidemiología y Salud Pública-CIBERESP, Instituto de Salud Carlos III, Madrid, Spain. [8]These authors contributed equally: Cristina Herencias, Laura Álvaro-Llorente. ✉e-mail: cherodr@gmail.com; asanmillan@cnb.csic.es; jeronimo.rodriguez.beltran@gmail.com

in *E. coli* collaterally increase susceptibility to β-lactams, fluoroquinolones, chloramphenicol, and tetracyclines[5]. Mecillinamresistant and tigecycline-resistant *E. coli* strains often show CS to nitrofurantoin[6]. In addition, mutations in horizontally transferred AMR genes (such as $bla_{CTX-M-15}$ and $bla_{OXA-48}$) that increase resistance towards modern β-lactams also produce CS to other, less modern β-lactams[7,8]. The emergence of CS is contingent on bacterial lifestyle[9], and it can be transiently induced using chemical compounds, suggesting that the physiological responses leading to CS can be exploited to tackle antibiotic-resistant infections[10].

We recently reported that the acquisition of plasmids isolated from clinical strains elicits CS[11]. Plasmids are arguably one of the most critical vehicles for AMR, as they readily spread through bacterial communities and typically carry multiple resistance determinants[12–15]. We found that plasmid-induced CS is robust and can be exploited to eliminate plasmid-containing clinical strains selectively. Interestingly, most plasmids tested in our previous study induced varying degrees of CS to azithromycin (AZI) and colistin (COL). This suggests that plasmid acquisition produces common physiological alterations that increase susceptibility to AZI and COL.

Here, we sought to identify the genetic determinants driving plasmid-associated CS. As a model, we first focus on the pOXA-48 plasmid. pOXA-48 is an enterobacterial, broad-host range, conjugative plasmid belonging to the plasmid taxonomic unit L/M that readily spreads in clinical settings[13]. Using a collection of clinically isolated pOXA-48 mutants, we first determine that the expression of $bla_{OXA-48}$ is solely responsible for inducing CS to AZI and COL. We next show that other clinically relevant plasmid-encoded β-lactamases produce similar CS responses in multiple, phylogenetically unrelated *E. coli* strains. Finally, by analysing resistance data comprising thousands of bacteria-antibiotic pairs, we demonstrate that β-lactamase-induced CS is a conserved phenomenon across different Enterobacterales. Altogether, our results pave the way for treatments that exploit β-lactam resistance as the Achilles' heel of multidrug-resistant enterobacteria.

## Results

### The β-lactamase $bla_{OXA-48}$ gene is responsible for pOXA-48-mediated CS

We first focused on the pOXA-48 plasmid, whose acquisition elicits CS to AZI and COL[11]. The pOXA-48 plasmid (~65 kb) encodes the β-lactamase $bla_{OXA-48}$ together with nearly 90 genes of diverse functions, all potentially contributing to the CS phenotype. To uncover the genetic determinant(s) responsible for plasmid-mediated CS, we took advantage of a well-characterised collection of seven pOXA-48 natural variants[16]. These plasmid variants carry diverse genetic mutations (deletions, insertions, and single nucleotide polymorphisms (SNPs)). These mutations affect critical plasmid functions such as conjugation, antibiotic resistance, or replication control and have been extensively characterised[16] (see Supplementary Fig. 1 and Supplementary Data 1). We introduced these plasmid variants in *E. coli* K12 strain J53 and determined their effect on the minimal inhibitory concentration (MIC) of four antibiotics: amoxicillin-clavulanic acid (AMC), ertapenem (ERT), azithromycin, and colistin (Fig. 1a). As expected, most plasmid variants conferred resistance to AMC and ERT (i.e. pOXA-48_WT, pOXA-48_PV-E, pOXA-48_PV-K, pOXA-48_PV-D, pOXA-48_PV-B, and pOXA-48_PV-O). All these six variants showed a reduced MIC of AZI and COL compared to the plasmid-free strain, confirming that the acquisition of pOXA-48 induces CS. Crucially, the pOXA-48_DEL variant, which carries a 199 bp deletion that abolishes $bla_{OXA-48}$ expression[16] (Supplementary Fig. 1 and Supplementary Data 1), neither conferred resistance to AMC and ERT nor showed increased susceptibility to AZI and COL (Fig. 1a).

To confirm the abolition of the CS phenotype, we quantified the number of viable cells for the pOXA-48_WT and pOXA-48_DEL carrying strains after exposure to different concentrations of COL and AZI. While the viability of the pOXA-48_WT-carrying strain significantly

dropped at low concentrations of AZI and COL (Mann-Whitney U test $p < 0.04$), the pOXA-48_DEL-carrying strain showed almost the same viability as the plasmid-free strain (Mann-Whitney U test $p > 0.33$ for all concentrations and both antibiotics; Fig. 1b, c). These results cannot be explained by differences in plasmid stability, fitness cost across plasmid variants (Supplementary Fig. 2), nor by differences in $bla_{OXA-48}$ expression levels across antibiotic treatments (Supplementary Fig. 3), and thus confirm that the pOXA-48_DEL variant, unlike pOXA-48_WT, does not produce CS to AZI or COL.

### The $bla_{OXA-48}$ gene is necessary and sufficient to induce CS to AZI and COL

The above results suggested that a fully functional $bla_{OXA-48}$ gene might be required to induce CS to AZI and COL. To test this hypothesis, we followed two complementary strategies. First, we used the pOXA-48_TET plasmid, a pOXA-48 version in which the $bla_{OXA-48}$ gene is replaced with a tetracycline resistance gene (*tetC*) (Fig. 2a and Supplementary Data 1). As expected, the pOXA-48_TET plasmid did not confer resistance to β-lactam antibiotics (ERT and AMC; Fig. 2b) and conferred resistance to tetracycline (Supplementary Fig. 4). In addition, pOXA-48_TET did not increase the susceptibility to AZI and COL, indicating that none of the genes in the plasmid backbone (nor the *tetC* gene) is responsible for the CS phenotype (Fig. 2b).

Second, we cloned the wild-type $bla_{OXA-48}$ gene under the control of its own promoter into the pUN4 plasmid[17], giving rise to the pUN-OXA-48_WT plasmid (Fig. 2a). As a control, we also cloned the same plasmid region amplified from the pOXA-48_DEL plasmid (pUN-OXA-48_DEL; Fig. 2a). We independently transformed these plasmids and the empty vector (pUN4) into *E. coli* and assessed the antibiotic susceptibility of the derivative strains (Fig. 2b). The empty pUN4 or the pUN-OXA-48_DEL plasmids did not significantly alter the susceptibility patterns compared to plasmid-free *E. coli* (Mann-Whitney U test $p > 0.28$). In contrast, production of the OXA-48 β-lactamase from the pUN4 plasmid (pUN-OXA-48_WT) conferred high resistance levels to AMC and ERT (Mann-Whitney U test $p < 0.004$) and increased susceptibility to AZI and COL (Mann-Whitney U test $p < 0.02$, Fig. 2b), qualitatively recapitulating the CS patterns of pOXA-48_WT. We note, however, that despite showing similar or even higher $bla_{OXA-48}$ expression levels (Supplementary Fig. 3), pUN-OXA-48_WT produced lower AMC resistance levels and a lower CS response to AZI (but not to COL) than pOXA-48_WT (Mann Whitney U test p = 0.0002; Fig. 2b). Nevertheless, these results indicate that a fully functional $bla_{OXA-48}$ gene is necessary and sufficient to induce CS to AZI and COL.

### CS to AZI and COL is a general side-effect of β-lactamase expression

The above results, together with our previous data showing that several β-lactamase-producing plasmids elicit varying degrees of CS to AZI and COL[11], prompted us to investigate whether the expression of other β-lactamases induces CS. To this end, we selected seven clinically relevant β-lactamases frequently encoded on mobile genetic elements. Specifically, we chose penicillinases (i.e. TEM-1), oxacillinases (OXA-1), extended-spectrum β-lactamases (CTX-M-15, CTX-M-14), metallo-β-lactamases (NDM-1), and other different carbapenemases (OXA-48, KPC-2) (Fig. 3a)[18–21]. We cloned these β-lactamases into the pBAD18 plasmid under the control of the arabinose-inducible $P_{BAD}$ promoter (Fig. 3a)[22]. We transformed these plasmids into *E. coli* K12 BW25113, a strain that cannot metabolise arabinose. This strain shows similar, although slightly lower, resistance levels than J53 (Supplementary Fig. 5). We then tested different arabinose concentrations to induce β-lactamase expression from the $P_{BAD}$ promoter. The arabinose concentration at which the resistance provided by the pBAD-OXA-48 plasmid matched the resistance provided by the natural pOXA-48_WT plasmid was chosen for further experiments with all β-lactamase-containing pBAD18 plasmids (Supplementary Fig. 6). This

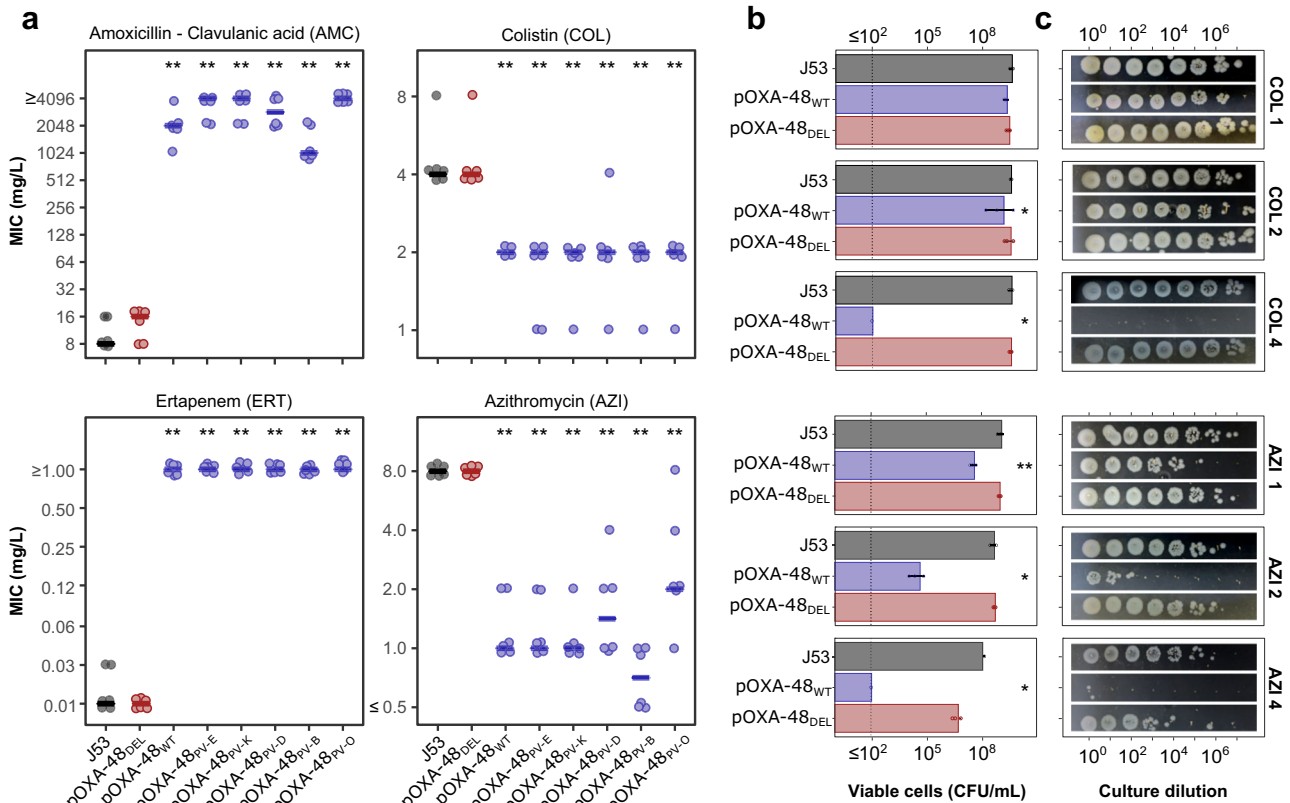

**Fig. 1 | A collection of pOXA-48 derivatives highlights that $bla_{OXA-48}$ expression is the most likely cause behind CS. a** MIC values of four antibiotics for *E. coli* J53 carrying the pOXA-48 variants. Horizontal lines indicate the median values of six biological replicates, represented as individual points. Strains carrying non-functional variants of the $bla_{OXA-48}$ gene are represented in red, strains with fully functional $bla_{OXA-48}$ gene versions in blue, and the plasmid-free wild-type strain in dark grey. Data points have been slightly jittered to avoid overlapping. AMC: amoxicillin-clavulanic acid, ERT: ertapenem, COL: colistin, and AZI: azithromycin. **b** Surviving viable cells after treatment with different concentrations of COL (upper panels) and AZI (lower panels). Data are presented as median values +/− standard deviation of three biological replicates (individual data points). The antibiotic and its concentration, in mg/L, are indicated on the right-hand side of each panel. The dashed vertical line depicts the detection limit of the experiment. **c** Spot test showing the survival at different antibiotic concentrations of *E. coli* J53 and its plasmid-carrying derivatives. After treatment with either AZI or COL at the indicated concentrations, appropriate dilutions of the surviving bacteria were plated on LB. Genotypes are shown in (**b**). In all panels, asterisks denote statistically significant differences compared to plasmid-free J53 (*$p < 0.05$, **$p < 0.01$; two-sided Mann-Whitney U test). Source data are provided in Source Data file.

concentration also maximizes the CS response to AZI and COL of the pBAD-OXA-48 plasmid (Supplementary Fig. 7). We note, however, that at this arabinose concentration, the expression of $bla_{OXA-48}$ from the $P_{BAD}$ promoter is smaller than that produced from its native promoter (Supplementary Fig. 3). We then measured the susceptibility patterns to 14 antibiotics from nine drug families for all β-lactamase-containing pBAD18 plasmids.

Under arabinose induction, all plasmids produced the expected β-lactam resistance profile according to their respective β-lactamases (Fig. 3b). Notably, all β-lactamases increased susceptibility to COL (albeit non-significantly; two-way ANOVA adjusted using Dunnett's test, $p > 0.12$ in all cases) and produced a substantial reduction in AZI resistance (two-way ANOVA adjusted using Dunnett's test, $p < 0.003$ in all cases). Importantly, heterologous expression of other genes (i.e. *gfp*, *cat*, and *tetA*) did not produce any significant increase in susceptibility to AZI or COL (Supplementary Fig. 8), indicating that the production of different β-lactamases (and not other proteins) induce CS to AZI and, to some extent, to COL.

### β-lactamase acquisition induces CS in diverse *E. coli* strains

We then decided to test whether the CS response elicited by β-lactamase expression is a general phenomenon that can be extrapolated to other *E. coli* strains. To assess the degree of conservation of β-lactamase-induced CS, we selected nine genetically diverse *E. coli* strains isolated from various hosts and geographical locations from

the ECOR collection[23,24]. We chose these strains because they lack known mobile β-lactamases, colistin and macrolide resistance genes and span all major *E. coli* phylogroups (Fig. 4a and Supplementary Data 2)[24,25]. *E. coli* wild-type strains are typically able to metabolise arabinose and are unsuitable for $P_{BAD}$-driven expression. We thus transformed these strains with the pUN4 empty vector and the pUN-OXA-48$_{WT}$ or pUN-CTX-M-15 plasmids. Analogous to the pUN-OXA-48$_{WT}$, the pUN-CTX-M-15 expresses the β-lactamase $bla_{CTX-M-15}$ under the control of its native promoter (Supplementary Data 1).

As expected, $bla_{OXA-48}$ or $bla_{CTX-M-15}$ provided resistance to ERT or cefotaxime (CTX), regardless of the initial resistance profile of the strain (Supplementary Fig. 9). Moreover, both β-lactamases produced a generalised increase in susceptibility to AZI and COL. Specifically, $bla_{CTX-M-15}$ led to significant CS to AZI and COL in five and four strains, respectively, while $bla_{OXA-48}$ significantly reduced the MIC of AZI and COL in five and four strains, respectively (Fig. 4b; Mann-Whitney U test $p < 0.038$, in all cases). Although only two strains showed significant CS to both AZI and COL (ECOR27 and ECOR49), virtually all strains (8 out of 9) showed CS to at least one of the antibiotics, indicating that β-lactamase-induced CS is a common phenomenon in *E. coli*.

### Detection of CS patterns in clinical MIC surveillance data

Given that the expression of horizontally acquired β-lactamases is the primary mechanism of β-lactam resistance in *E. coli*[26], we then hypothesised that a negative correlation between β-lactam resistance and

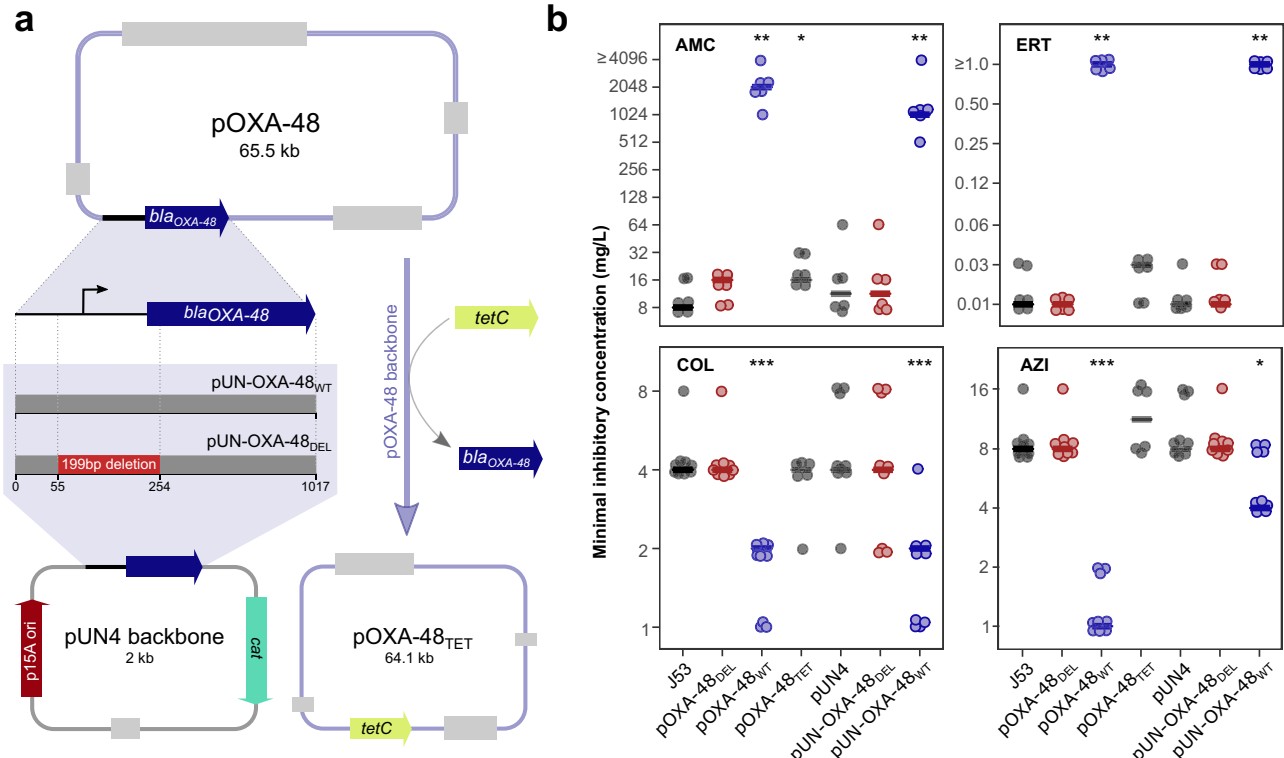

**Fig. 2 | The expression of $bla_{OXA-48}$ induces collateral sensitivity. a** Scheme depicting the design and construction of the pUN4 derivative vectors and the pOXA-48$_{TET}$. Left branch: the region encoding the $bla_{OXA-48}$ gene was amplified from the pOXA-48$_{WT}$ plasmid and cloned into the pUN4 backbone, obtaining the pUN-OXA-48$_{WT}$ vector. A parallel procedure was carried out for the pOXA-48$_{DEL}$ plasmid, leading to the pUN-OXA-48$_{DEL}$ vector. Right branch: the region encoding the *tetC* gene was amplified and cloned into the backbone of the pOXA-48$_{WT}$ plasmid, replacing $bla_{OXA-48}$ and resulting in the pOXA-48$_{TET}$ plasmid. **b** MIC determination for *E. coli* J53 carrying the pOXA-48$_{TET}$ plasmid and the pUN4 derivatives in four antibiotics: AMC: amoxicillin-clavulanic acid, ERT: ertapenem, COL:

colistin, and AZI: azithromycin. Horizontal lines indicate the median values and each datapoint represents a biological replicate ( n = 6 for ERT and AMC; n = 9 for AZI and COL except for pOXA-48$_{TET}$, in which n = 6). Strains carrying non-functional variants of the $bla_{OXA-48}$ gene are represented in red, strains with fully functional $bla_{OXA-48}$ gene versions in blue, and β-lactamase-free strains in dark grey. Data points have been slightly jittered to avoid overlapping. Asterisks denote statistically different susceptibility levels compared to plasmid-free J53 (*$p < 0.05$, **$p < 0.01$, ***$p < 0.001$; two-sided Mann-Whitney U test). Source data are provided as a Source Data file.

resistance to COL or AZI would be consistent with β-lactamase-induced CS. To test the validity of this assumption, we searched for significant anticorrelations in our experimental dataset. We combined all experimental data from Figs. 1–4 and found that resistance to β-lactam antibiotics is indeed significantly anti-correlated with COL and AZI resistance (Spearman rho < −0.27; $p < 0.04$ in all cases), even after excluding non-β-lactamase producers from the analyses (Fig. 5a, Supplementary Fig. 10 and Supplementary Fig. 11 and Supplementary Data 3), indicating that correlation analysis can detect CS signatures across different strains and mechanisms of resistance.

To further investigate the signature of β-lactamase-associated CS in clinical enterobacteria, we analysed the Antimicrobial Testing Leadership and Surveillance (ATLAS) database. ATLAS comprises antibiotic susceptibility data for 44 antimicrobials and more than 600,000 pathogens belonging to more than 200 bacterial species isolated from over 70 countries[27,28]. For each bacterial isolate, ATLAS stores MIC data of multiple antibiotics (mode: 11, range: 11-34), which allows the detection of associations of resistance patterns between antibiotic pairs[29,30].

We computed all possible pairwise correlations between the MIC data of 23 antibiotics for the ~80,000 *E. coli* isolates stored in ATLAS. Analysis of ~500 antibiotic pairs revealed three weak yet significant negative associations. All of them occurred between β-lactam antibiotics and COL. Specifically, strains showing higher resistance levels to carbapenem antibiotics doripenem (DOR), meropenem (MER), and imipenem (IMP) tend to show higher susceptibility to COL (i.e. are

negatively correlated; Spearman's rho −0.12, −0.14, −0.19, respectively, and Bonferroni adjusted $p < 0.027$ in all cases; Fig. 5a, b and Supplementary Data 3). Our experimental dataset further supports this result, as both analyses identified significant anticorrelations between IMP and MER resistance and COL (we did not measure DOR resistance experimentally), highlighting that carbapenem resistance elicits CS to COL.

To gain a deeper insight into this result, we analysed the MIC distributions of COL, grouping the bacteria based on their respective carbapenem resistance levels. As expected, strains with higher levels of carbapenem resistance tend to show significantly higher COL susceptibility (Supplementary Fig. 12), supporting the idea that the observed anticorrelations are a signature of β-lactamase-induced CS rather than a result of other potential confounding factors. It is important to note that despite being increasingly used to treat Gram-negative infections[31], AZI susceptibility is rarely tested in *E. coli*, and thus, the low number of AZI susceptibility determinations in ATLAS restricted our analysis to COL.

We then decided to test if the signature of β-lactamase-induced CS was detectable in other enterobacteria. To this end, we analysed MIC data for the most abundant species within the order Enterobacterales in the ATLAS database. We found negative correlations between COL and β-lactam antibiotics for 13 out of 16 pathogens in a species- and antibiotic-specific pattern (Fig. 5c and Supplementary Fig. 13). The specificity of the CS signature agrees with previous reports showing that the effect of antibiotic combinations is species-specific

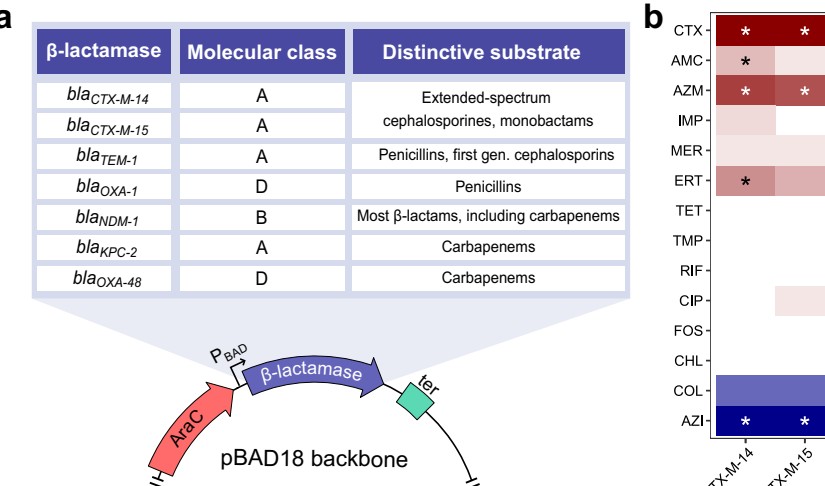

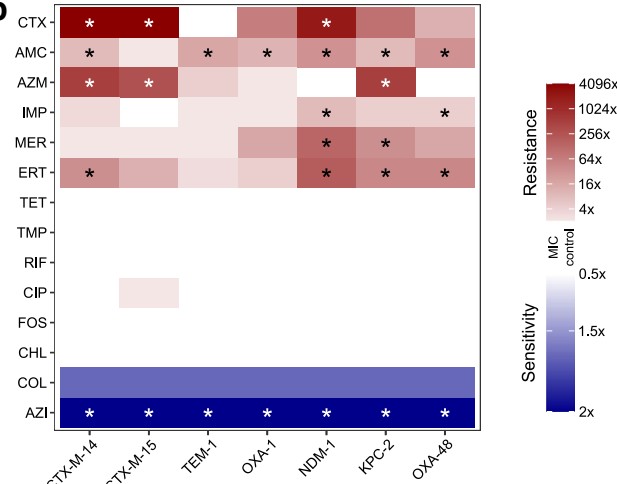

**Fig. 3 | Production of different β-lactamases induces CS to AZI and, to some extent, to COL. a** The molecular class and distinctive substrate of each β-lactamase are listed in the table. β-lactamases were cloned under the control of the P_BAD promoter in the pBAD18 plasmid. **b** Heat map representing collateral responses to antibiotics associated with β-lactamase acquisition (median of six biological replicates). The colour code represents the fold change in MIC of each antibiotic for the β-lactamase-carrying derivatives relative to *E. coli* BW25113 strain carrying pBAD18 without β-lactamase. Asterisks denote statistically significant differences (two-way ANOVA followed by Dunnett's test, adjusted $p < 0.047$ in all cases). CTX, cefotaxime; AMC, amoxicillin-clavulanic acid; AZM, aztreonam; IMP, imipenem; MER, meropenem; ERT, ertapenem; TET, tetracycline; TMP, trimethoprim; RIF, rifampicin; CIP, ciprofloxacin; FOS, fosfomycin; CHL, chloramphenicol; COL, colistin; and AZI, azithromycin. Source data are provided in Source Data file.

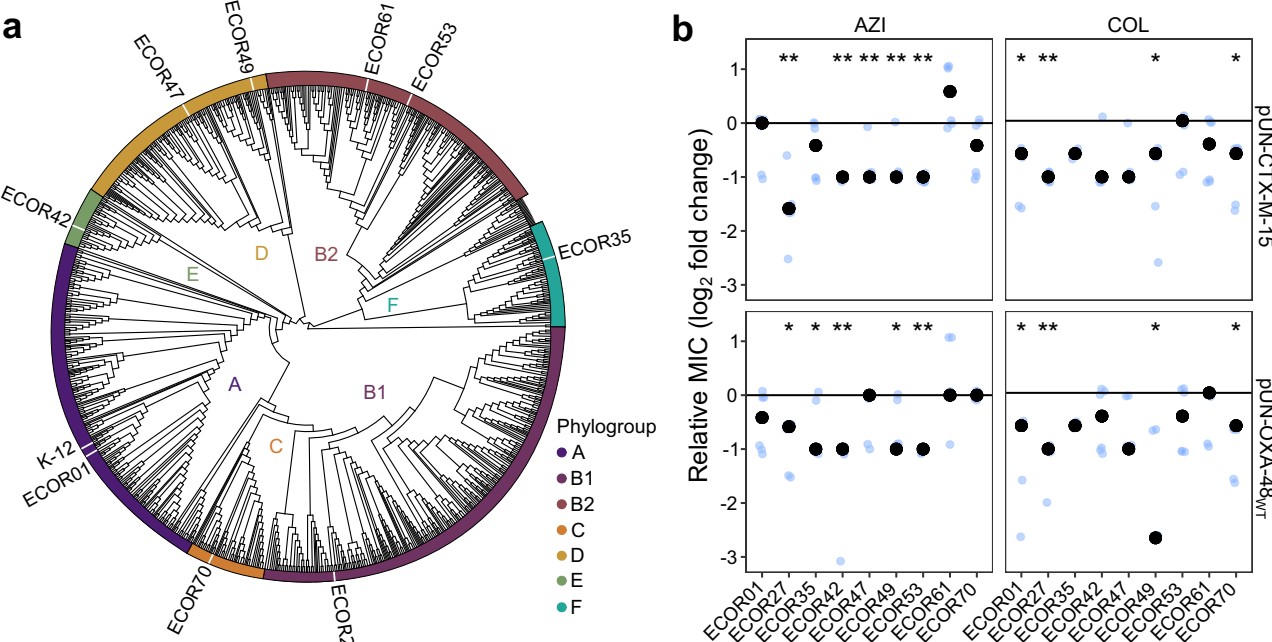

**Fig. 4 | Collateral sensitivity induced by β-lactamases is conserved across phylogenetically distant *E. coli* strains. a** Phylogenetic tree depicting the relationships of 1573 representative *E. coli* genomes and highlighting the strains used in this study. Colours at the tip of tree branches represent different phylogroups, as indicated in the legend. **b** Collateral sensitivity response to azithromycin (AZI) and colistin (COL) associated with the presence of active β-lactamases in *E. coli*. Relative MIC (in log2 fold-change) was determined as the MIC of the β-lactamase-carrying *E. coli* relative to the β-lactamase-free strain carrying the empty vector. Black points represent the median values of six biological replicates, shown as individual blue points. The horizontal line represents no change in MIC relative to the control strain. Asterisks denote a significant decrease in MIC (*$p < 0.05$, **$p < 0.01$; two-sided Mann-Whitney U test). The data points have been slightly jittered to avoid overlapping. Source data are provided in Source Data file.

and that β-lactamase expression produces host-specific physiological effects[32,33]. Altogether, these results suggest that β-lactamase-induced CS might be pervasive across enterobacteria despite the vast diversity of intrinsic susceptibility patterns and resistance mechanisms.

## Discussion

In this study, we uncovered that different, clinically relevant β-lactamases increase bacterial susceptibility to AZI and COL. By taking a classical genetics approach, we first demonstrated that the *bla*_OXA-48

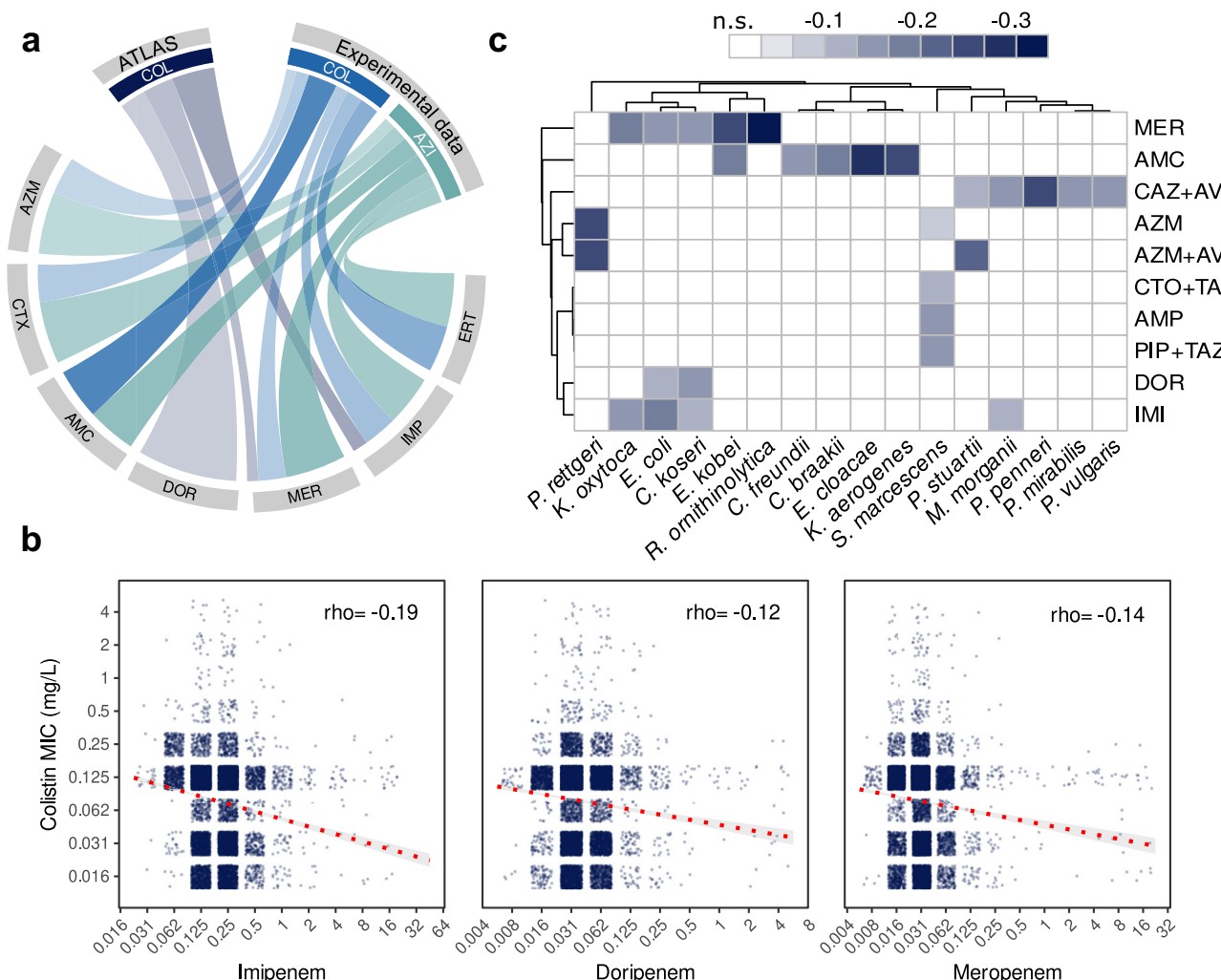

**Fig. 5 | Antibiotic resistance surveillance data show patterns consistent with β-lactamase-induced CS. a** The plot displays negative correlations between ATLAS COL (dark blue) or experimental AZI or COL MIC data (light blue and green, respectively) and β-lactam antibiotics using experimental data or data from ATLAS. The strength of the colours is proportional to the strength of the correlation (Spearman's rho). Only statistically significant correlations are plotted (Spearman correlation Bonferroni adjusted $p < 0.05$). **b** MICs for COL and the three antibiotics (imipenem, doripenem, meropenem, in mg/L) showing significant anticorrelations in ATLAS data. Each dot within the plots represents an isolate, and the red dotted line represents the best linear fit with 95% confidence intervals in light grey. Spearman's rho is also indicated within each plot. Bonferroni adjusted $p < 0.0001$ in all cases. **c** Hierarchical heatmap revealing negative correlations between β-lactam resistance and COL in different species belonging to the Enterobacterales order. Only statistically significant negative correlations were included. The strength of the colours is proportional to the strength of the correlation (Spearman's rho). Antibiotic pairs with no significant correlation are denoted by white, and negative correlation between antibiotics by blue colour as shown in the legend. MER meropenem, AMC amoxicillin-clavulanic acid, CAZ + AVI: ceftazidime + avibactam, AZM aztreonam, AZM + AVI: aztreonam + avibactam, CTO + TAZ ceftolozane + tazobactam, AMP: ampicillin, PIP + TAZ: piperacillin + tazobactam, DOR doripenem, IMI: imipenem. Source data are provided in Source Data file.

carbapenemase is solely responsible for the CS response to COL and AZI induced by the acquisition of the broadly disseminated pOXA-48 plasmid (Figs. 1 and 2). Given that all β-lactamases share a common mechanism of action (i.e. hydrolysis of the β-lactam ring in the periplasm), we tested whether this result could be generalised to other β-lactamases and strains. Using heterologous expression systems, we discovered that most β-lactamases induce CS to AZI and COL and that this phenomenon is conserved in phylogenetically diverse *E. coli* strains (Figs. 3 and 4). We then searched for signatures of β-lactamase-induced CS in a clinical surveillance database containing millions of MIC data for different pathogens. We found significant negative correlations between MICs to different β-lactam antibiotics and COL resistance in *E. coli* and other Enterobacterales, which strongly supports the notion of β-lactamase-induced CS (Fig. 5).

Our results indicate that acquiring β-lactamases of different families and classes induces a common pleiotropic response that

increases susceptibility to AZI and COL in *E. coli*. But what is the molecular mechanism underlying this phenotype? A compelling hypothesis might come from the observation that β-lactamase expression affects bacterial physiology, reducing fitness in different species[16,22,34–37]. In some cases, this effect has been traced to the β-lactamase signal peptide, suggesting that protein translocation or accumulation in the periplasm perturbs cellular homeostasis[22,33,35,38,39]. Supporting this idea, some β-lactamases have been shown to induce the cellular envelope stress response[33,38]. In other cases, the biological cost occurs as a byproduct of residual dd-peptidase activity of certain β-lactamases. This ancestral activity may decrease cross-linked muropeptides, affecting peptidoglycan composition and producing structural changes in the cell wall[36,40]. Altogether, this evidence suggests that β-lactamase accumulation may destabilise the bacterial envelope, facilitating AZI and COL activity. Accordingly, AZI permeability and COL activity have been shown to increase when the

envelope integrity is compromised[41–44]. In line with this hypothesis and our results, expression of the $bla_{VIM-2}$ metallo-β-lactamase has been recently shown to induce the envelope stress response and disrupt the integrity of the outer membrane, rendering the bacteria more susceptible to AZI in both *E. coli* and *Klebsiella pneumoniae*[45]. Future work addressing this phenomenon will shed light on the molecular mechanisms driving β-lactamase-induced CS, with the potential reward of uncovering the ultimate mechanisms responsible for β-lactamase fitness costs, which have remained elusive for over 20 years.

β-lactams are the most prescribed antibiotics, and resistance rates are skyrocketing among clinical pathogens[46]. The Centers for Disease Control and Prevention (CDC) and the World Health Organization (WHO) consider β-lactamase-producing Gram-negative bacteria among the world's most serious threats[47,48]. This has led to a worldwide effort to potentiate the activity of existing β-lactam antibiotics by developing new β-lactamase inhibitors, among other strategies[49–52]. In this regard, our results pave the way for novel therapies that exploit CS using COL and AZI in combination with β-lactam antibiotics to eliminate β-lactamase-producing bacteria. We note, however, that the magnitude of the CS response in our experiments is moderate (2-8-fold reductions in MIC). While this result is similar to the CS responses found elsewhere[4,53,54], more research is needed before β-lactamase-induced CS can be exploited therapeutically. The rational design of novel derivatives of macrolides and polymyxins[55–57] or novel antibiotic combinations[32,58] hold promise for increasing the efficacy of treatments against β-lactamase-producing bacteria. In summary, future work will be needed to understand the molecular mechanism(s) behind β-lactamase-induced CS and to find or design compounds that exploit it efficiently.

## Methods

### Bacterial strain and culture conditions

*E. coli* strains were routinely grown in Lennox lysogeny broth (LB) at 37 °C with shaking (225 rpm) or in agar plates (15 g/L). When needed, kanamycin (50 mg/L), carbenicillin (100 mg/L), cefotaxime (50 mg/L), tetracycline (15 mg/L), or chloramphenicol (30 mg/L) were added for plasmid selection. Supplementary Data 4 lists all antibiotics used in this study and their sources. *E. coli* J53 (F⁻ met pro Azi$^r$; ref. [59]) was used as the recipient of the pOXA-48 plasmid variants in conjugation experiments[16]. For consistency, we used this strain in all experiments except those involving the induction of the $P_{BAD}$ system. J53 readily metabolises L-arabinose (Sigma Aldrich, Cat. A3256), and therefore, for those experiments, we used *E. coli* strain BW25113 (F⁻ DE(*araD-araB*) 567 *lacZ*4787(del)::*rrnB*-3 LAM⁻ *rph*-1 DE(*rhaD-rhaB*)568 *hsdR*514; ref. [60]). Both strains (J53 and BW25113) are *E. coli* K12 derivative strains and present similar levels of resistance and susceptibility to antibiotics (Supplementary Fig. 4). *E. coli* strains belonging to each major phylogroup were chosen from the ECOR collection[23] to represent the genetic diversity of *E. coli*. The presence of known β-lactamases other than the chromosomally-encoded *ampC* gene present in all *E. coli* strains was ruled out using deposited genomic sequences (Bioproject PRJNA230969 and ref. [24]) and Resfinder[61] (Supplementary Data 2).

### Plasmid construction

All plasmids and oligonucleotides used in the study are listed in Supplementary Data 1 and Supplementary Data 5, respectively. To construct the β-lactamase-carrying pUN4 plasmids (p15A origin of replication, chloramphenicol resistance), standard molecular biology techniques in combination with the Gibson assembly method were used[62,63]. PCR primer pairs were designed to amplify the pUN4 plasmid backbone (CS25-CS26, CS7-CS8 or CS41-CS42). PCR primer pairs were also designed to amplify the $bla_{OXA-48}$ gene containing the upstream region from pOXA-48$_{WT}$ or pOXA-48$_{DEL}$ plasmids (CS23-CS24 and CS9-CS10) and the $bla_{CTX-M-15}$ gene from pBAD-CTX-M-15 (CS43 and CS44). The resulting DNA fragments were cloned using the Gibson Assembly

Cloning Kit (NEB, Cat. E5510S) into plasmid backbone pUN4 to obtain pUN-OXA-48$_{WT}$, pUN-OXA-48$_{DEL}$, and pUN-CTX-M-15, respectively. The final plasmid constructs were introduced into *E. coli* DH5α and were confirmed by PCR and whole plasmid sequencing (Accession numbers: PP735911, PP735912, PP735909, and PP735914).

Prof. Linus Sandegren provided arabinose-inducible pBAD18 (pMB1 origin of replication, kanamycin resistance) plasmids carrying the β-lactamase genes used in this work. Non-β-lactamase genes used as controls were cloned into the pBAD18 using the Gibson Assembly Cloning Kit (NEB, Cat. E5510S). pBAD (RBS)_fwd/rev primer pairs were used to amplify the plasmid backbone. Primer pairs (Supplementary Data 5) were designed to amplify *cat* and *gfp* genes from pBGC vector[64] and the *tetA* gene from the pCT plasmid[65]. The final plasmid constructs were introduced into *E. coli* DH5α and the transconjugant strains were confirmed by PCR and whole plasmid sequencing (Accession numbers: PP735913, PP735915, PP735910, PP735916). To create pOXA-48$_{TET}$, the $bla_{OXA-48}$ gene was replaced by the tetracycline resistance gene (*tetC*) by homologous recombination[66]. The gene *tetC* with its own promoter was amplified from pACYC184 using primers oxa4tet_rpl_f(r) with 40 bp homologous regions directly up and downstream of the $bla_{OXA-48}$ coding sequence. Gene replacement was verified by PCR using primers oxa48_out_f(r) (Supplementary Data 5) and whole plasmid sequencing (Accession number: PP735908).

### Bacterial growth analysis

Single colonies of each bacterial population were inoculated in LB starter cultures and incubated at 37 °C for 16 h at 225 rpm (six biological replicates). Each culture was diluted 1:2000 in LB, and 200 μl was added to a 96-well microtiter plate containing appropriate antibiotic concentrations. When appropriate, L-arabinose (0, 0.005, 0.1, 0.2, or 0.5% (w/v)) was added into the culture broth to induce gene expression in the pBAD18-carrying strains. For checkerboard analysis, a matrix in 96-well plates was created with 2-fold serial dilutions of arabinose and amoxicillin-clavulanic acid. Plates were incubated for 22 h at 37 °C with strong orbital shaking before reading the optical density at 600 nm (OD600) every 15 min in a Synergy HTX (BioTek) plate reader. The area under the growth curve (AUC) was obtained using the 'auc' function from the 'flux' R package. We used AUC as it integrates all relevant growth parameters (maximum growth rate, lag duration and carrying capacity). For GFP expression monitoring, after 22 h of incubation, the fluorescent emission was measured at 528 nm with excitation at 485 nm using a fluorescence spectrophotometer Synergy HTX (BioTek). The result is shown as relative fluorescence to the optical density at 600 nm. Data was represented using an R custom script and the 'ggplot2' package.

### Plasmid stability

To test the stability of pOXA-48-like plasmids after each antibiotic treatment, the surviving bacterial populations were collected from the well corresponding to 0.5x MIC of either COL or AZI. Serial dilutions ranging from $10^{-1}$ to $10^{-7}$ were plated on both LB agar and LB agar with carbenicillin to identify and count the pOXA-48$_{WT}$-carrying colonies (including negative controls of plasmid-free wild-type clones). For pOXA-48$_{DEL}$, PCR amplification with the oligonucleotides OXA-F and OXA-R or ΔblaOXA-48F and ΔblaOXA-48R (Supplementary Data 5) was used to assess plasmid stability of 100 independent colonies.

### Antimicrobial susceptibility testing

The antibiotic minimal inhibitory concentration (MIC) was determined using broth microdilution tests in LB medium, as reported before[11] (see Supplementary Fig. 14 for a comparison between MIC results in LB and MHII). Briefly, starter cultures were prepared and incubated as described above. Each culture was diluted 1:2000 in LB, resulting in approximately ~5·$10^4$ colony forming units, and 200 μl were added to a 96-well microtiter plate containing the appropriate antibiotic

concentration. Optical density at 600 nm (OD600) was measured after 22 h of incubation at 37 °C with strong orbital shaking every 10 min in a Synergy HTX (BioTek) plate reader to determine MIC values. The MIC value is reported as the lowest antibiotic concentration, resulting in no visible growth (measured here as OD600 < 0.2)[11]. The MICs of amoxicillin + clavulanic acid (AMC) and fosfomycin (FOS) were determined using the agar dilution method. Each starter culture was diluted 1:1000 in LB medium, and 5 μl was plated on LB agar containing the appropriate antibiotic concentration. LB agar plates were supplemented with glucose-6-phosphate (25 mg/L, Sigma-Aldrich, Cat 10127647001) for fosfomycin MIC determination. Similarly to the broth microdilution assays, 0.5% (w/v) arabinose was added when necessary. To ensure reproducibility, we performed parallel MIC determinations of plasmid-free J53 control strain in every assay.

To perform disk-diffusion assays, sterile disks containing the antibiotic were placed on plates of LB previously swabbed with a 0.5 McFarland matched bacterial suspension and then incubated at 37 °C for 22 h. Inhibition halos were measured using ImageJ software. Antibiotic disk content used in the assays were azithromycin 15 μg, colistin 10 μg, cefotaxime 30 μg and amoxicillin + clavulanic acid 30 μg (20 μg + 10 μg; all from Bio-Rad).

### Determination of $bla_{OXA-48}$ expression levels
To determine the expression levels of $bla_{OXA-48}$ after antibiotic treatment, single colonies of each bacterial population were inoculated in LB starter cultures and incubated at 37 °C for 16 h at 225 rpm (three biological replicates). Each culture was diluted 1:2000 in LB and LB with colistin (0.25 mg/L), azithromycin (0.25 mg/L), ertapenem (0.125 mg/L or 0.0015 mg/L in the case of non-induced pBAD-OXA-48), or arabinose at 0.5% w/v as appropriate. The cultures were incubated at 37 °C for 16 h at 225 rpm, and cells were collected by centrifugation (3 min, 10,000 × g) and stored at −80 °C. RNA extraction was performed with the Total RNA Purification Kit (Norgen Biotek Corp, Cat. 17200) following the manufacturer's instructions. Reverse transcription was carried out with 1 μg of RNA as a template and the High-Capacity cDNA Reverse Transcription Kit (Applied Biosystems, Cat. 10400745) following the manufacturer's instructions. To quantify expression levels of $bla_{OXA-48}$ and $rpoB$ (endogenous control), 2 μL of diluted cDNA (1:10) was used as a template in a 20 μL-reaction containing NZYSupreme qPCR Green Master Mix (Nzytech, Cat. MB41901), and specific primers pairs for each gene (Supplementary Data 5). The efficiencies of each reaction were evaluated using the standard curve method in triplicate ($bla_{OXA-48}$: Efficiency = 1.02, $R^2$ = 0.99; $rpoB$: Efficiency = 1.02, $R^2$ = 0.99). Fluorescence data was measured in a 7500 Real-Time PCR System (Applied Biosystems). Relative mRNA quantities were calculated using the comparative threshold cycle (Ct) method and normalized using $rpoB$ gene expression (ΔCt). The gene expression fold change (FC) values were calculated by the ΔΔCt method ($2^{-ΔΔCt}$) considering as control the BW25113-pBAD-OXA-48 group grown in LB without Arabinose. Three biological replicates were quantified in triplicate (technical replicates).

### Plasmid sequencing
The pOXA-48$_{TET}$ plasmid was extracted using the Wizard genomic DNA purification kit (Promega, Cat. A2360). The pUN4 and pBAD18 derivatives were isolated using the Plasmid Easypure isolation kit (Macherey-Nagel, Cat. 740727.50) following the manufacturer's protocol. DNA was quantified using the Qubit dsDNA Assay (Thermo-Fisher, Cat. Q32851) following the manufacturer's instructions. Genomic and plasmid DNA were sequenced with Oxford Nanopore technology at Plasmidsaurus (https://www.plasmidsaurus.com/, Eugene, EEUU) using the V10 chemistry library prep kit with the R10.4.1 flow cells. Sequences were annotated using the pLannotate software[67] and analysed using the BLAST Ring Image Generator[68].

### Phylogenetic analysis of *E. coli* strains
To determine the distribution of the *E. coli* isolates across the phylogeny of the species, we obtained a total of 1573 *E. coli* genomes from the RefSeq database (https://www.ncbi.nlm.nih.gov/assembly), including all isolates from the ECOR Collection[23] in.fna format. Genomes were assigned to an *E. coli* phylogroup using the EzClermont tool (https://github.com/nickp60/EzClermont)[69]. Genomic distances were estimated using Mash[70], and a phylogenetic tree was constructed using the R package PATO[71].

### ATLAS data processing
To conduct our analysis, we gathered MIC data for all species belonging to the order Enterobacterales in the ATLAS database (www.atlas-surveillance.com). To avoid possible bias due to susceptibility testing heterogeneity, we discarded pathogen-antibiotic pairs with less than 50 instances. We also updated the classification of *Enterobacter aerogenes* to *Klebsiella aerogenes* to align with recent taxonomic changes[72]. In ATLAS, the actual MIC is sometimes reported as below or equal (≤) the detection limit of the susceptibility test or exceeds (>) the highest antibiotic concentration tested. In these cases, we made conservative approximations. If the MIC was lower or equal to the lowest concentration tested, we recorded it as equal to that concentration. Conversely, if the MIC exceeded the highest concentration tested, we recorded it as double the detection limit. For instance, if ATLAS raw data indicated ">64 mg/L", we used "128 mg/L" as the numerical value. We computed pairwise correlations among the log2-transformed numerical MIC data for all possible antibiotic pairs for each species. This analysis was performed using the "corr.test" function from the "psych" package in R, with the Spearman method applied. We specifically focused on identifying significant negative correlations in our analysis. To visualise these correlations, we utilised the "circlize" package for chord diagrams and the "pheatmap" package for heatmaps in R.

### Statistics
Data sets were analysed using R software version 4.1.2. Normality was assessed by visual inspection and the Shapiro-Wilk test. ANOVA tests were performed to ascertain the effect of the 'β-lactamase x antibiotic concentration' interaction term in the analysis of minimal inhibitory concentration. Plasmid stability was explored using the chi-square test. Associations between antibiotic resistance were performed using a simple linear model. Mann-Whitney U test was used to assess the significance of MIC determinations for the CS instances in natural *E. coli* strains.

### Reporting summary
Further information on research design is available in the Nature Portfolio Reporting Summary linked to this article.

## Data availability
Datasets generated and/or analysed during the current study are included in the Source Data file with this paper and can be downloaded from the following repository[73]: https://doi.org/10.5281/zenodo.11061421. ATLAS data can be visualised through the ATLAS website (https://atlas-surveillance.com) and was downloaded from the dataset published in ref. 74 Plasmid sequences can be accessed under Genbank accession numbers: PP735908, PP735911, PP735912, PP735909, PP735914, PP735913, PP735910, PP735915, and PP735916 (see also Supplementary Data 1). ECOR whole genome sequences are available under BioProject accession number PRJNA230969 Source data are provided with this paper.

## Code availability
The source code used to produce the results and analyses presented in this manuscript is available from ref. 73, https://github.com/JeroRB/Collateral_Sensitivity or https://doi.org/10.5281/zenodo.11061421.

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

## Acknowledgements

We are indebted to Prof. Jesús Blázquez (CNB-CSIC, Spain) for sharing the ECOR strains used in this study. We thank Prof. Linus Sandegren (Uppsala University, Sweden), Christopher Frøhlich (The Artic University of Norway, Norway), and Martin Palm (University of Gothenburg, Sweden) for kindly sharing the pBAD-*bla* plasmids, the pUN4 vector and the pOXA-48$_{TET}$, respectively. Work in the Evodynamics lab (https://evodynamicslab.com/) is supported by project no. PI21/01363 (J.R-B) and PI23/01945 (C.H), funded by the Carlos III Health Institute (ISCIII) and co-funded by the European Union; CIBER -Consorcio Centro de Investigación Biomédica en Red- (CIBERINFEC) CB21/13/00084 (J.R-B, C.H); Convocatoria SEIMC-FUNDACIÓN SORIA MELGUIZO de Investigación 2021 (J.R-B); and funded by the European Union (ERC, HorizonGT, 101077809; J.R-B). Views and opinions expressed are however those of the author(s) only and do not necessarily reflect those of the European Union or the European Research Council Executive Agency. Neither the European Union nor the granting authority can be held responsible for them. The work at A.S.M lab was supported by the European Research Council (ERC) under the European Union's Horizon 2020 research and innovation programme (ERC grant no.757440-PLASREVOLUTION). CH is supported by a Sara Borrell contract from the Carlos III Health Institute (ISCIII) (grant no. CD21/00115) and the Convocatoria Intramural Emergentes 2021 FIBioHRC-IRYCIS. Cod. IPM-21 nº C13. J.R-B acknowledges support by a Miguel Servet contract from the Carlos III Health Institute (ISCIII) (grant no. CP20/00154), co-founded by the European Social Fund, 'Investing in your future'. L.A.L is granted by the Comunidad de Madrid through the Consejería de Educación, Universidades, Ciencia y Portavocía (grant no. PEJ-2021-AI/BMD-23127). P.R.M is a recipient of a predoctoral PFIS grant (grant no. FI22/00265) from the Carlos III Health Institute (ISCIII) through the Recovery, Transformation and Resilience Plan and Next Generation EU from the Europen Union. A.F-C was funded by MCIN/AEI/10.13039/501100011033 and by the 'European Union NextGenerationEU/PRTR' (Grant FJC2021-046751-I).

## Author contributions

C.H., L.Á.-L., A.F.-C., A.M.-C.; J.D.F., and L.J-S. performed experiments; C.H., L.Á.-L., A.F.-C,. A.M.-C., P.R.-M., J.R.-B. analysed the data and made the figures. F.E.G. constructed the pOXA-48$_{TET}$ plasmid; R.C., J.A.C.-P. gave technical support and conceptual advice. C.H., Á.S.M, and J.R.-B. conceived and supervised the project and wrote the paper with input from all the authors. All authors discussed and provided critical feedback during the analysis of results and manuscript writing at all stages.

## Competing interests

The authors declare no competing interests.
