## [Peer Review File · Nature Communications]

REVIEWER COMMENTS

Reviewer #1 (Remarks to the Author):

In the manuscript “Beta-lactamase expression induces collateral sensitivity in *Escherichia coli*”, Herencias et al. investigated the effect of beta-lactamase expression on collateral sensitivity to other antibiotic classes such as the protein synthesis inhibitor azithromycin and the bacterial cell envelope-targeting colistin. As the spread of beta-lactam resistance genes via horizontal gene transfer led to beta-lactam resistant bacteria becoming one of the major health care threats nowadays, exploring new ways of treating beta-lactam resistant infections is indispensable. In this study the authors examined how the expression of beta-lactamases encoded on plasmids induces collateral sensitivity toward other antibiotics. For this purpose, naturally spreading plasmids were used for measuring resistance, and were further cloned in order to prove that the expression of blaOXA-48 is necessary and sufficient to induce collateral sensitivity to colistin (COL) and azithromycin (AZI). Moreover, it was proven that beta-lactamases from diverse molecular classes also lead to collateral sensitivity in phylogenetically distant *E. coli* strains. Extending the study to the Antimicrobial Testing Leadership and Surveillance (ATLAS) database, the authors also found albeit weak negative correlations between MICs to different β -lactam antibiotics and COL resistance in *E. coli* and other Enterobacterales. In the discussion, the authors provided a possible explanation to this phenomenon, namely that the accumulation of the expressed β -lactamase in the periplasm might destabilise the bacterial envelope, leading to sensitivity to antibiotics from other antibiotic classes. Overall, this study describes an important phenomenon that can pave the way towards novel therapies to cope with beta-lactam resistant infections, proving statements by convincing experimental evidences. However, I missed some additional experiments that would substantially strengthen their findings that the expression level of the beta-lactamase genes in the presence of AZI and COL are indeed responsible for the observed CS phenotypes.

Major points:

The expression of the beta-lactamase gene is usually tightly controlled and turned on only in the presence of beta-lactams in line with the authors’ explanation that beta-lactamase affects bacterial physiology and reduces fitness in different species. It would be interesting to see whether the beta-lactamase genes are indeed also expressed in the presence of AZI and COL (from their native promoter; e.g. does the blaOXA48 gene show increased expression from its native promoter in the presence of COL and AZI?)

I am wondering about the correlation between the AZI and COL minimal inhibitory concentration (MIC) and increasing arabinose dosages in case of the strains carrying the pBAD_betaLactamase variants. One would expect a negative correlation if the increased expression of the beta-lactamase genes reduces the MIC of the strains to AZI and COL.

The COL and AZI MIC results on Figure 2b show a high variability in case of the pUN-OXA48_wt plasmid. I am wondering what can explain this phenomenon. Have you tested the stability of this plasmid compared to the empty vector in the absence or presence of different antibiotics?

On Figure 4b the Authors state that “collateral sensitivity response to AZI and COL is associated with beta-lactamase expression”. However the authors have not measured the expression rate of the beta-lactamase genes. It would be interesting to see that the beta-lactamase expression indeed correlates to the strength of collateral sensitivity. For example, on some strain backgrounds where the expression rate of the beta-lactamase is high are the ones showing the highest level of collateral sensitivity to COL and AZI.

Did the authors correct the p values for multiple testing (such as FDR correction) during the correlation tests between the ~500 antibiotic pairs taken from the ATLAS dataset (line #259-265; Figure 5a).

On Supplementary Figure 8 would the authors still see the significant negative correlation if they only include the beta-lactamase carrying strains?

On Figure 4b what is the authors’ explanation about the large variability between the MIC of the biological replicates?

Minor points:

In Figure 1c, I recommend increasing the contrast of the spot test photos.

On Supplementary Figure 2a was the fitness measured in the absence or in the presence of an antibiotic? It should be clearly indicated in the figure legends.

What was the statistical test used on Figure 2b? In case of the pUN-OXA48_wt only 2 of the 6 replicates showed reduced MIC compared to the empty plasmid. I think it would be more accurate to indicate the strength of the p values such as: * $p < 0.05$ ** $p < 0.1$ *** $p < 0.001$.

Did you clone the TET resistance gene under its own promoter into the pOXA-48_wt plasmid?

On line # 167-168 it would be more accurate to state that: - The resistance level provided by the OXA48 was fine-tuned from the Pbad promoter to match the strain carrying the natural pOXA-48wt plasmid- rather than their current sentence: “We then fine-tuned expression levels to match those found in natural plasmids using blaOXA-48 as a reference” as the exact level of gene expression was not compared among them. Was the same arabinose concentration used during the experiments with the additional beta-lactamase variants as well?

What were the selectable marker genes on the pUN4 and on the pBAD18 empty vectors for the transformation?

On Figure 5a what the colors represent exactly? It is not clear from the figure legends.

Reviewer #2 (Remarks to the Author):

In this manuscript, the authors search to better understand the collateral sensitivity (CS) mechanisms associated to beta-lactamases carried in plasmid. From previous experiments the authors identified that beta-lactamase mediate resistance concomitantly leads to increased sensitivity to other antibiotics, azithromycin and colistin, specifically. The authors thoroughly and systematically study the dependencies of the CS effect on the presence, expression and dependence in genetic background. The results presented in this study are significant and represent a significant step to establish an effective framework for the rationale of using CS as a strategy to mitigate the spread of antibiotic resistant pathogens. The evaluations in genetically different phylogroups of *E. coli* and in clinically relevant strains is of particular relevance. The manuscript is well-written and the results are presented very effectively in well conceived and clear figures. I only have very minor comments and look forward to see this manuscript in Nature Communications.

Minor Comments:

- In the introduction reference 13 is not superscript.
- In figure 1, there is some variability within the population carrying some of the plasmid. More strongly seen against AZI in pOXA-48PV-O. Is this due to plasmid stability due to the different SNPs or INDELS?
- Also, in the same figure 1A it is a little inconsistent how all of the replicates of the deletion, and even J53 a little bit, are growing at AZI 8 but in the viability assay they show reduced cell viable numbers even at AZI 2. Can you comment this discrepancy?

Reviewer #3 (Remarks to the Author):

The manuscript “ β -lactamase expression induces collateral sensitivity in *Escherichia coli*” describes an approach to elucidate the phenomenon of collateral sensitivity produced by expression of beta-lactamases in *E. coli*. The authors follow up on their recent publication of this phenomenon and convincingly show that the CS phenotype is indeed attributable to the expression of individual beta-lactamases, although the magnitude of the effect differs and is usually modest (as many CS interactions presented before). The authors clearly broaden the knowledge by showing that the effect is general to many clinically relevant beta-lactamases and that it can be replicated in several genetic backgrounds, although to varying levels, as can be expected. Overall, the findings are interesting and add to the growing amount of data on the negative effects of beta-lactamase expression for bacterial fitness, and open up for more extensive analysis of the possibilities for designing therapeutic regimes that utilize the CS phenotype to limit resistance development.

The most convincing results are those with the pOXA-48 plasmid, where the authors clearly show that the CS phenotype is attributable to the functional OXA-48. After that, the results are less clear, perhaps due to expression differences from the different plasmids and expression systems (see comments below). I believe the data and that the CS effect is real, but the way the authors have tested this and sometimes present the data makes it less convincing in its current state and could be improved.

My main concerns are listed below.

Lines 150-154 and Figure 2b. The AZI MIC for the cloned pUN-OXA-48wt is clearly different from the pOXA-48wt. 3/5 replicates show the same MIC as the strains lacking OXA-48, and the other two just one dilution step lower which is usually regarded as natural variation. I am surprised that this would be statistically different from the nonfunctional OXA-48. Still, the authors state that the susceptibility to AZI is increased and recapitulates that of pOXA-48wt (Lines 152-154). Could the weaker effect be due to a difference in the expression of OXA-48 from the pUN-plasmid? The MIC of AMC for pUN-OXA-48wt is also slightly lower than for pOXA-48 in agreement with that.

Figure 3b. It is very hard to appreciate the differences in MIC between the different beta-lactamases the way it is presented here. The scales for the heatmap are very hard to read with very different ranges, for example light red is 2-4x while medium red is 4-256x. Maybe it would make more sense to include actual MICs or make a more fine-graded heat map?

For the pOXA-48 plasmid, the CS for AZI was 8-fold while all beta-lactamases produce 1.5-2-fold CS when expressed from pBAD. Is this an effect of the difference in susceptibility in BW25113 seen in Supplementary Figure 4, even though the authors state that the strains have comparable MICs (Line 166), or is it an effect of a lower expression level of the beta-lactamases? Since the expression level of beta-lactamases is directly correlated with the resistance level and, correspondingly, should be correlated with the CS effect coming from any negative effect of the beta-lactamase activity, it is surprising that the authors did not test how a larger range of expression through higher arabinose concentrations affect the phenotype. Since the native expression from pOXA-48 gave the most reliable data on MICs and CS effects, it would also have been interesting to see how other native plasmids (such as the ones used in the author's previous publication) would behave in controlled experiments with/without their beta-lactamases. I do, however, realize that this would include quite some construction work.

Lines 202-204 and Supplementary Figure 7. The MICs to Cefotaxime are surprisingly low for CTX-M-15, with more than half of the strains having MICs below 1. The CTX-M-15 beta-lactamase usually gives MICs >32. The scale of the cefotaxime graph makes it impossible to determine the MICs higher than 1.

Figure 4. How was the MIC tested here? The distribution of the data points is very strange since MIC is determined in discrete steps. As a non-clinician, I can appreciate the use of half-steps for research purposes, but this looks like averages and not medians as stated in the figure legend. The scale ending at 0.0 (with data points at that position) is also strange for a relative MIC.

Lines 227-232 and Supplementary Figure 8. The designation of the beta-lactamase-free and beta-lactamase-containing strains must be wrong. Now, it indicates that beta-lactamase-free strains (blue) have higher MICs to beta-lactams and lower to COL and AZI.

Lines 269-274 and Supplementary Figure 9. The analyses made are only in the MIC range up to 1 m/L for Imipenem and 0.25 mg/L for Doripenem and Meropenem although the MICs for beta-lactamase-producing strains should be much higher. The clinical breakpoints, according to EUCAST, are 4, 2, and 8

for the respective antibiotics. So, if the CS was really associated with the production of beta-lactamases the authors should use a wider range of MICs to include those isolates and get a predicted stronger negative correlation at higher beta-lactam MICs.

Lines 397-412. Was there a specific reason why the authors used LB medium for their MIC determinations instead of the standardized MHII medium? MHII is usually more reproducible.

REVIEWER COMMENTS

Reviewer #1 (Remarks to the Author):

In the manuscript "Beta-lactamase expression induces collateral sensitivity in Escherichia coli", Herencias et al. investigated the effect of beta-lactamase expression on collateral sensitivity to other antibiotic classes such as the protein synthesis inhibitor azithromycin and the bacterial cell envelope-targeting colistin. As the spread of beta-lactam resistance genes via horizontal gene transfer led to beta-lactam resistant bacteria becoming one of the major health care threats nowadays, exploring new ways of treating beta-lactam resistant infections is indispensable. In this study the authors examined how the expression of beta-lactamases encoded on plasmids induces collateral sensitivity toward other antibiotics. For this purpose, naturally spreading plasmids were used for measuring resistance, and were further cloned in order to prove that the expression of blaOXA-48 is necessary and sufficient to induce collateral sensitivity to colistin (COL) and azithromycin (AZI). Moreover, it was proven that beta-lactamases from diverse molecular classes also lead to collateral sensitivity in phylogenetically distant E. coli strains. Extending the study to the Antimicrobial Testing Leadership and Surveillance (ATLAS) database, the authors also found albeit weak negative correlations between MICs to different β -lactam antibiotics and COL resistance in E. coli and other Enterobacterales. In the discussion, the authors provided a possible explanation to this phenomenon, namely that the accumulation of the expressed β -lactamase in the periplasm might destabilise the bacterial envelope, leading to sensitivity to antibiotics from other antibiotic classes. Overall, this study describes an important phenomenon that can pave the way towards novel therapies to cope with beta-lactam resistant infections, proving statements by convincing experimental evidences. However, I missed some additional experiments that would substantially strengthen their findings that the expression level of the beta-lactamase genes in the presence of AZI and COL are indeed responsible for the observed CS phenotypes.

We thank the reviewer for these helpful comments.

Major points:

The expression of the beta-lactamase gene is usually tightly controlled and turned on only in the presence of beta-lactams in line with the authors' explanation that beta-lactamase affects bacterial physiology and reduces fitness in different species. It would be interesting to see whether the beta-lactamase genes are indeed also expressed in the presence of AZI and COL (from their native promoter; e.g. does the blaOXA48 gene show increased expression from its native promoter in the presence of COL and AZI?)

We thank the reviewer for raising this issue and agree that differences in β -lactamase expression in the presence of different antibiotic treatments might affect our conclusions. To test this possibility, we have now performed RT-qPCR assays to directly assess the transcriptional level of *bla*_{OXA-48} from the strains and conditions tested through the paper. The

results show that bla_{OXA-48} is expressed from its native promoter at similarly high levels in all conditions tested. Indeed, statistical analysis shows that bla_{OXA-48} expression levels do not change across antibiotic treatments (ANOVA effect of *treatment* x *genotype* interaction $df=3$, $F=1.053$, $p=0.423$). This indicates that bla_{OXA-48} expression from its native promoter (pOXA-48_{WT} background) is constitutive rather than inducible, at least in our experimental conditions. These new results are shown in the new Supplementary Figure 3, and presented in lines 105, 155, and 178 in results and lines 441-462 in methods sections.

I am wondering about the correlation between the AZI and COL minimal inhibitory concentration (MIC) and increasing arabinose dosages in case of the strains carrying the pBAD_betaLactamase variants. One would expect a negative correlation if the increased expression of the beta-lactamase genes reduces the MIC of the strains to AZI and COL.

We agree with the reviewer that lower β -lactamase levels should reduce the strength of the CS response, and this should be observable with the pBAD-based expression system. To test this possibility, we have measured AZI and COL inhibition halos using disk-diffusion assays at different arabinose concentrations (see Supplementary Figure 7 reproduced below for convenience). The results show that, indeed, arabinose levels correlate with AZI and COL sensitivity (Spearman's rank correlation $\rho=0.87$ and 0.62 , respectively, and $P<0.02$ for both) and AMC resistance (Spearman's rank correlation $\rho=-0.95$, $P<10^{-6}$) for the strain carrying the pBAD-OXA-48 plasmid. Together with the data shown in Supplementary Figures 10 and 11, we believe that these results convincingly support the notion that the strength of the CS response is dependent on β -lactamase levels.

The COL and AZI MIC results on Figure 2b show a high variability in case of the pUN-OXA48_wt plasmid. I am wondering what can explain this phenomenon. Have you tested the stability of this plasmid compared to the empty vector in the absence or presence of different antibiotics?

The differences in one dilution observed in AZI and COL MIC for the pUN-OXA-48wt in Figure 2b can likely be attributed to multiple factors. First, it's important to consider the variability associated with biological replicates. Each data point represents an independent biological replicate. Small genetic differences accumulated during the experiment can thus slightly affect bacterial physiology, translating into slight differences in bacterial growth¹ that might influence the outcome of the MIC assay. Second, small changes in the assay conditions (pH, incubation time, media, etc.) due to inter-day variability can slightly change the MIC of many agents, including COL and AZI. In addition, we used different batches of

antibiotics across the replicates, which added another layer of variability. Finally, even under optimal conditions, MIC tests yield discontinuous results on a logarithmic scale, falling within 2-fold dilution intervals. Therefore, small differences often translate into 2-fold MIC changes. Even though these sources of variability are widely recognized in the field²⁻⁵, we performed three additional biological replicates for COL and AZI MIC determinations to show that our results are indeed reproducible within the assay variation (see new version of figure 2b). In addition, we can rule out that plasmid stability explains the variability observed. Indeed, both the pUN-OXA48wt and the empty vector are not significantly lost in none of the tested conditions. We have now included this information in the manuscript (See below and Figure Supp 2)

On Figure 4b the Authors state that “collateral sensitivity response to AZI and COL is associated with beta-lactamase expression”. However, the authors have not measured the expression rate of the beta-lactamase genes. It would be interesting to see that the beta-lactamase expression indeed correlates to the strength of collateral sensitivity. For example, on some strain backgrounds where the expression rate of the beta-lactamase is high are the ones showing the highest level of collateral sensitivity to COL and AZI.

The reviewer raises an interesting point. In the new version of the manuscript, we have performed RT-qPCR experiments to measure β -lactamase expression, as suggested (Supp Fig. 3). However, we argue that the level of available protein in the periplasm (i.e. active β -lactamase) is the biologically relevant parameter in our experiments. Indeed, β -lactamase expression has been traditionally quantified using colourimetric approaches that directly measure the catalytic activity of the β -lactamase, not its mRNA levels. For this reason, and because β -lactamase expression is strongly correlated with resistance levels⁶⁻¹⁰, we believe that measuring β -lactamase activity (i.e. resistance levels) is a better proxy for protein abundance than mRNA levels. Indeed, Supplementary Figures 7, 10 and 11 show that high levels of β -lactam resistance lead to higher susceptibility to COL/AZI, indicating that the strength of CS correlates with the levels of active β -lactamase within the cell and presumably with its expression.

However, after reviewing the paper, we realized that the term “expression” was probably overused throughout the manuscript and that this may lead to confusion, as we do not directly measure mRNA expression in many cases. For this reason, we have revised the

manuscript and removed the word “expression” or changed it to more general terms such as “ β -lactamase presence” or “production of β -lactamase” in the cases where expression was not directly measured (for instance, see lines 30, 72, 123, 153, 198, 321, 326). However, we refrain from changing the article's title, as we believe we have clearly linked β -lactamase expression (through measuring mRNA and resistance levels) with the CS response through the manuscript.

Did the authors correct the p values for multiple testing (such as FDR correction) during the correlation tests between the ~500 antibiotic pairs taken from the ATLAS dataset (line #259-265; Figure 5a).

The P values were corrected by multiple comparisons using Bonferroni's method. We thank the reviewer for pointing out that this info was not included in the previous version of the manuscript. We have now included it in the appropriate sections of the manuscript (Figure 5 legend L259 and L269)

On Supplementary Figure 8 would the authors still see the significant negative correlation if they only include the beta-lactamase carrying strains?

The reviewer raises an interesting point. After performing the suggested analyses, we found that excluding data from strains not containing functional β -lactamases (that is, those carrying either empty plasmids or *bla*OXA-48_{DEL}), some of the correlations are lost, which is to be expected as CS is likely explained by the presence (or absence) of a β -lactamase. However, even after removing non- β -lactamase producers, we still see significant negative correlations for 8 out of the 12 original antibiotic combinations (Spearman's correlation $\rho < -0.201$; $p < 0.02$ in all cases; see the plot below), indicating that even within resistant strains, variations in β -lactam resistance levels affect the strength of the CS response.

We now refer to this information in the manuscript (L239-243) and in a new Supplementary Figure (Supp. Fig 11; shown below)

On Figure 4b what is the authors' explanation about the large variability between the MIC of the biological replicates?

As we exposed above, several biological, physicochemical, and technical factors might explain variability in MIC data. We refer the reviewer to the previous response on this topic for an explanation.

Minor points:

In Figure 1c, I recommend increasing the contrast of the spot test photos

We have now increased the contrast of the spot tests.

On Supplementary Figure 2a was the fitness measured in the absence or in the presence of an antibiotic? It should be clearly indicated in the figure legends.

Fitness was measured in the absence of antibiotics. This is now included in the figure legend (Line 37; Supp. Material)

What was the statistical test used on Figure 2b?

Mann-Whitney U test was used to assess statistical significance. This was indicated in the text of the original manuscript. We have now specified the statistical test in the figure legend as well.

*In case of the pUN-OXA48_wt only 2 of the 6 replicates showed reduced MIC compared to the empty plasmid. I think it would be more accurate to indicate the strength of the p values such as: * $p < 0.05$ ** $p < 0.1$ *** $p < 0.001$.*

Following this and other comments from Reviewer's #2 and #3, we decided to perform three additional AZI and COL MIC measurements (biological replicates) for all strains in Figure 2b (except for pOXA_{TET}) to confirm the consistency of the CS phenomenon. They have been included in the figure and the statistical analysis. We also have modified the figure to represent the strength of the p values, as suggested by the reviewer.

Did you clone the TET resistance gene under its own promoter into the pOXA-48_wt plasmid?

Yes, the pOXA-48_{TET} plasmid was created by replacing the *bla*_{OXA-48} gene and its promoter sequence with the tetracycline resistance gene (*tetC*) with its own promoter. We have added a detailed description of the pOXA-48_{TET} plasmid in the methods section to clarify this issue (Lines 389-394)

On line # 167-168 it would be more accurate to state that: - The resistance level provided by the OXA48 was fine-tuned from the P_{bad} promoter to match the strain carrying the natural pOXA-48wt plasmid- rather than their current sentence: "We then fine-tuned expression levels to match those found in natural plasmids using blaOXA-48 as a reference" as the exact level of gene expression was not compared among them.

We agree with the referee that gene expression was not measured in the original manuscript. However, we argue that the level of available protein in the periplasm (i.e. active β -lactamase) is actually the biologically relevant parameter in our experiments. For this reason, and because β -lactamase expression is strongly correlated with resistance levels⁶⁻¹⁰, we believe that measuring resistance is a better proxy for protein abundance than gene expression. To clarify this and in agreement with the reviewer's suggestion, we have changed the sentence to read as follows:

"We then tested different arabinose concentrations to induce β -lactamase expression from the P_{BAD} promoter. The arabinose concentration at which the resistance provided by the pBAD18-bla_{OXA-48} plasmid matched the resistance provided by the natural pOXA-48_{WT} plasmid was chosen for further experiments with all β -lactamase-containing pBAD18 plasmids" (Line 172-180)

Was the same arabinose concentration used during the experiments with the additional beta-lactamase variants as well?

Yes, we have now included this information in the main manuscript (see previous response).

What were the selectable marker genes on the pUN4 and on the pBAD18 empty vectors for the transformation?

pUN4 and the pBAD18 confer chloramphenicol and kanamycin resistance, respectively. This information was initially presented in the Supplementary Table 1. We have now included information about replication origin and resistance in the description of the vectors in the Material and Methods section (Lines 372 and 382).

On Figure 5a what the colors represent exactly? It is not clear from the figure legends.

The colours were meant to help the reader identify the connections between ATLAS COL (dark blue), Experimental COL (light blue), and Experimental Azi (green) resistance levels and β -lactam resistance levels. We have now included a sentence in the figure legend to clarify this point (Lines 265-267).

Reviewer #2 (Remarks to the Author):

In this manuscript, the authors search to better understand the collateral sensitivity (CS) mechanisms associated to beta-lactamases carried in plasmid. From previous experiments the authors identified that beta-lactamase mediate resistance concomitantly leads to increased sensitivity to other antibiotics, azhitromycin and colistin, specifically. The authors

thoroughly and systematically study the dependencies of the CS effect on the presence, expression and dependence in genetic background. The results presented in this study are significant and represent a significant step to establish an effective framework for the rationale of using CS as a strategy to mitigate the spread of antibiotic resistant pathogens. The evaluations in genetically different phylogroups of E. coli and in clinically relevant strains is of particular relevance. The manuscript is well-written and the results are presented very effectively in well conceived and clear figures. I only have very minor comments and look forward to see this manuscript in Nature Communications.

We thank the reviewer for their positive comments.

Minor Comments:

- In the introduction reference 13 is not superscript.

Changed. We thank the referee for spotting this oversight.

- In figure 1, there is some variability within the population carrying some of the plasmid. More strongly seen against AZI in pOXA-48PV-O. Is this due to plasmid stability due to the different SNPs or INDELS?

We thank the reviewer for highlighting this point. As we explained above, the variability in replicates of MIC results is inherent to the technique used. This phenomenon is commonly recognized in the field of antimicrobial susceptibility testing²⁻⁵.

Regarding the influence of plasmid stability in the CS response, we have demonstrated that the pOXA-48_{PV-O} plasmid is 100% stable under the evaluated conditions, as evidenced in Supplementary Figure 2. However, it is true that the different SNPs and INDELS in the plasmid backbone might influence bacterial physiology, therefore affecting the strength of the CS.

- Also, in the same figure 1A it is a little inconsistent how all of the replicates of the deletion, and even J53 a little bit, are growing at AZI 8 but in the viability assay they show reduced cell viable numbers even at AZI 2. Can you comment this discrepancy?

We may have failed to understand the reviewer's comment, but we think that the discrepancies observed are to be expected due to the different nature of each technique represented. Figure 1A illustrates the minimal inhibitory concentration (MIC) values for each strain, denoting the lowest concentration of the antimicrobial agent (in mg/L) that inhibits visible bacterial growth. For example, for strains J53 and pOXA-48_{DEL}, the MIC for AZI is 8 mg/L, indicating their inability to proliferate at this concentration. Consequently, the highest AZI concentration at which viable cells are observable is 4 mg/L. This matches Figures 1B and 1C, showing that J53 and the deletion mutant grow at this concentration but not pOXA_{WT}, whose MIC is 1 mg/L.

Reviewer #3 (Remarks to the Author):

The manuscript “β-lactamase expression induces collateral sensitivity in Escherichia coli” describes an approach to elucidate the phenomenon of collateral sensitivity produced by expression of beta-lactamases in E. coli. The authors follow up on their recent publication of this phenomenon and convincingly show that the CS phenotype is indeed attributable to the expression of individual beta-lactamases, although the magnitude of the effect differs and is usually modest (as many CS interactions presented before). The authors clearly broaden the knowledge by showing that the effect is general to many clinically relevant beta-lactamases and that it can be replicated in several genetic backgrounds, although to varying levels, as can be expected. Overall, the findings are interesting and add to the growing amount of data on the negative effects of beta-lactamase expression for bacterial fitness, and open up for more extensive analysis of the possibilities for designing therapeutic regimes that utilize the CS phenotype to limit resistance development.

The most convincing results are those with the pOXA-48 plasmid, where the authors clearly show that the CS phenotype is attributable to the functional OXA-48. After that, the results are less clear, perhaps due to expression differences from the different plasmids and expression systems (see comments below). I believe the data and that the CS effect is real, but the way the authors have tested this and sometimes present the data makes it less convincing in its current state and could be improved.

My main concerns are listed below.

We thank the reviewer for the positive comments and thorough evaluation of our manuscript.

Lines 150-154 and Figure 2b. The AZI MIC for the cloned pUN-OXA-48wt is clearly different from the pOXA-48wt. 3/5 replicates show the same MIC as the strains lacking OXA-48, and the other two just one dilution step lower which is usually regarded as natural variation. I am surprised that this would be statistically different from the nonfunctional OXA-48. Still, the authors state that the susceptibility to AZI is increased and recapitulates that of pOXA-48wt (Lines 152-154). Could the weaker effect be due to a difference in the expression of OXA-48 from the pUN-plasmid? The MIC of AMC for pUN-OXA-48wt is also slightly lower than for pOXA-48 in agreement with that.

Thank you for bringing this point up. We have conducted additional analysis and experiments to address this comment and Reviewer's #1 concern about variability in MIC measurement. Specifically:

1.) We performed COL and AZI MIC assays for three additional biological replicates of the experiments shown in Figure 2b (up to a total of n=9). The results demonstrate that the observed phenomenon is reproducible and that expression of *bla*_{OXA-48} from pUN4 leads to a significant reduction of AZI MIC compared to the J53 MIC. However, the reviewer is right that the strength of the CS response is lower than that found for the pOXA-48wt plasmid. We now note this in the manuscript, where we indicate the following:

Line 153: *“In contrast, production of the OXA-48 β-lactamase from the pUN4 plasmid (pUN-OXA-48_{WT}) conferred high resistance levels to AMC and ERT (Mann-Whitney U test $p < 0.004$) and increased susceptibility to AZI and COL (Mann-Whitney U test $p < 0.02$, **Fig. 2b**),*

qualitatively recapitulating the CS patterns of pOXA-48_{WT}. We note, however, that despite showing similar or even higher bla_{OXA-48} expression levels (Supplementary Fig. 3), pUN-OXA-48_{WT} produced lower AMC resistance levels and a lower CS response to AZI (but not to COL) than pOXA-48_{WT} (Mann Whitney U test $p = 0.0002$; Fig. 2b)”

2.) To ascertain the possible causes of this difference, we measured bla_{OXA-48} expression levels through RT-qPCR. The results, shown in the new Supplementary Fig. 3, indicate that bla_{OXA-48} is expressed at higher levels from the pUN vector than from pOXA-48wt. This is somewhat expected due to the higher copy number of pUN4 (10-15 copies per cell). However, despite showing lower bla_{OXA-48} expression levels, pOXA-48wt confers a higher AMC resistance level than pUN-OXA-48. This indicates that the levels of active protein within the periplasm, rather than gene expression, determine the strength of the CS response. This notion is supported by the correlations between β -lactam resistance and CS to AZI or COL shown in Supplementary Fig. 10 and 11.

We are currently investigating the mechanisms behind the relatively high pOXA-48wt resistance levels for a different project. Our data (still preliminary) suggest that an additional gene located in the pOXA-48 backbone might contribute to OXA-48 protein stability and/or translocation to the periplasm, increasing the ratio of active protein per mRNA and, thus, resistance levels.

Figure 3b. It is very hard to appreciate the differences in MIC between the different beta-lactamases the way it is presented here. The scales for the heatmap are very hard to read with very different ranges, for example light red is 2-4x while medium red is 4-256x. Maybe it would make more sense to include actual MICs or make a more fine-graded heat map?

The figure in the original version of the manuscript was designed to emphasize CS rather than resistance. However, we acknowledge that the scale was a bit hard to read, so we changed it to better reflect the resistance levels (see new Figure 3).

For the pOXA-48 plasmid, the CS for AZI was 8-fold while all beta-lactamases produce 1.5-2-fold CS when expressed from pBAD. Is this an effect of the difference in susceptibility in BW25113 seen in Supplementary Figure 4, even though the authors state that the strains have comparable MICs (Line 166), or is it an effect of a lower expression level of the beta-lactamases?

We agree with the Reviewer that the strength of the CS response shown in Figure 4 is smaller than that shown for pOXA-48wt. We believe that this might be due to the combination of both inter-strain differences and a lower expression level. As the reviewer points out, pOXA-48wt confers lower resistance levels to β -lactam antibiotics in BW25113 than in J53 (see Supp. Figure 5). This matches recent observations that show that pOXA-48wt-mediated resistance levels vary among strains¹¹, and suggest that strain-specific variations might slightly affect resistance levels, and, consequently, the strength of the CS response. We now acknowledge this small difference in resistance levels in (L170): “We transformed these plasmids into *E. coli* K12 BW25113, a strain that cannot metabolise arabinose. This strain shows similar, although slightly lower resistance levels compared to J53”

In addition, we have now quantified bla_{OXA-48} expression levels from the pOXA-48 plasmid

and the heterologous expression systems used in the paper (pUN and pBAD). This information is depicted in the new Supp. Figure 3. Results show that, indeed, β -lactamase expression, as well as the β -lactam resistance level, from the pBAD vector is smaller than that observed from the pOXA-48, albeit slightly. This difference in gene expression might also explain why the strength of the observed CS is smaller. We have clarified this in the new version of the manuscript as follows: (Line 178) “We note, however, that at this arabinose concentration, the expression of bla_{OXA-48} from the P_{BAD} promoter is smaller than that produced from its native promoter (Supplementary Fig. 3)”.

In summary, strain-specific conditions and lower expression levels from the P_{BAD} promoter translate into lower active β -lactamase levels in the periplasm and, consequently, a smaller CS effect.

Since the expression level of beta-lactamases is directly correlated with the resistance level and, correspondingly, should be correlated with the CS effect coming from any negative effect of the beta-lactamase activity, it is surprising that the authors did not test how a larger range of expression through higher arabinose concentrations affect the phenotype.

We agree with the Reviewer that expression, resistance, and CS should positively correlate, and therefore, higher expression should produce higher CS levels. This is indeed what is shown in the new Supp. Figure 7. However, the $araC$ - P_{BAD} promoter produces relatively low expression levels compared to other inducible systems, even at maximum induction¹². Moreover, the concentration we used is already relatively high (between 2-5 fold greater than what is typically used), and the P_{BAD} promoter is known to be saturated at this concentration¹²⁻¹⁵. For these reasons, we did not test arabinose concentrations above 0.5% (w/v).

Since the native expression from pOXA-48 gave the most reliable data on MICs and CS effects, it would also have been interesting to see how other native plasmids (such as the ones used in the author's previous publication) would behave in controlled experiments with/without their beta-lactamases. I do, however, realize that this would include quite some construction work.

We agree with the reviewer that measuring the effect of β -lactamase expression on CS using natural plasmids could be interesting. However, we decided not to perform the suggested experiments mainly for the following two reasons: *i)* Most natural plasmids carry more than one antibiotic-resistance gene (often several β -lactamases) and many other genes whose expression may lead to unpredictable phenotypic effects, complicating the interpretation of results. For this reason, we initially focused on pOXA-48, a plasmid that we¹⁶⁻¹⁹ and others²⁰⁻²⁵ have extensively characterised. pOXA-48 carries a single resistance gene (bla_{OXA-48}), and we already had a natural variant without the β -lactamase (pOXA-48_{DEL}). *ii)* In the manuscript, we already show experimental results with variants of a natural plasmid with and without β -lactamases (pOXA48_{DEL}, pOXA48_{TET}), two different synthetic vector backbones (pUN4 and pBAD), two laboratory strains (BW25113 and J53), seven different β -lactamases, and ten natural strains (ECOR). Moreover, we further validated the results with a completely different approach (ATLAS). Therefore, we believe that the notion that β -lactamase production leads to CS is already well supported in the manuscript. However, we appreciate the reviewer's comment, and we plan to use a CRISPRi approach to analyze the

potential change in CS profiles associated with natural plasmids when we silence β -lactamase expression.

Lines 202-204 and Supplementary Figure 7. The MICs to Cefotaxime are surprisingly low for CTX-M-15, with more than half of the strains having MICs below 1. The CTX-M-15 beta-lactamase usually gives MICs >32. The scale of the cefotaxime graph makes it impossible to determine the MICs higher than 1.

After reading the reviewer's comment, we were also a bit puzzled by the low values of CTX and ERT MICs. For this reason, we decided to repeat the experiment, this time using a higher concentration of CTX and ERT (32 and 4 mg/L, respectively) as the maximum tested value. The results, represented in Supplementary Figure 9 and reproduced below for convenience, show that, indeed, CTX-M-15 and OXA-48 confer high-level resistance to CTX and ERT (≥ 64 mg/L and >1 mg/L, respectively, for most strains).

Figure 4. How was the MIC tested here? The distribution of the data points is very strange since MIC is determined in discrete steps. As a non-clinician, I can appreciate the use of half-steps for research purposes, but this looks like averages and not medians as stated in the figure legend. The scale ending at 0.0 (with data points at that position) is also strange for a relative MIC.

We agree with the reviewer that the previous version of Figure 4 was a bit confusing. The figure shows the MIC values of the β -lactamase producing strains (ECOR/pUN-bla) relative to the strain carrying the empty plasmid (ECOR/pUN4). This was indicated in the axis title as "Relative MIC". However, the axis scale was perhaps a bit confusing because, as the referee points out, some data points seemed to be at 0. We have corrected this issue by changing the axis to a logarithmic scale. The new version of Figure 4 (shown below for convenience) now shows log₂ Fold Change in Relative MIC, which can also be interpreted as the number of dilution steps that the MIC decreases (or increases) in the presence of a β -lactamase. We have changed the legend accordingly.

Lines 227-232 and Supplementary Figure 8. The designation of the beta-lactamase-free and beta-lactamase-containing strains must be wrong. Now, it indicates that beta-lactamase-free strains (blue) have higher MICs to beta-lactams and lower to COL and AZI.

Thank you for spotting this oversight. We have changed the figure legend accordingly.

Lines 269-274 and Supplementary Figure 9. The analyses made are only in the MIC range up to 1 m/L for Imipenem and 0.25 mg/L for Doripenem and Meropenem although the MICs for beta-lactamase-producing strains should be much higher. The clinical breakpoints, according to EUCAST, are 4, 2, and 8 for the respective antibiotics. So, if the CS was really associated with the production of beta-lactamases the authors should use a wider range of MICs to include those isolates and get a predicted stronger negative correlation at higher beta-lactam MICs.

We thank the reviewer for bringing this to our attention. In the previous version of Supplementary Figure 9 (now Supp. Figure 12), MIC distributions for highly carbapenem resistant strains were absent from the plot. This was because each carbapenem MIC had few strains in ATLAS (i.e. $n < 50$) to draw representative distributions and significance analyses. We have now corrected this by merging together all carbapenem MIC groups showing less than 50 strains, which correspond to carbapenem-resistant strains (i.e. imipenem MIC ≥ 2 and meropenem and doripenem MICs ≥ 0.5 mg/L). We have also revisited the figure to show significant differences more clearly. This has been changed in the new version of Supp. Figure 12, which is shown below for convenience.

We also note that strains showing high carbapenem MICs were never excluded in the correlation analyses shown in Figure 5 (note the few data points at high carbapenem MICs in Fig. 5b). We also note that many carbapenemases, such as blaOXA-48 (and other β -lactamases)²⁶⁻²⁹, confer low levels of resistance to carbapenems that do not typically reach the EUCAST breakpoints.

Lines 397-412. Was there a specific reason why the authors used LB medium for their MIC determinations instead of the standardized MHII medium? MHII is usually more reproducible.

We agree with the reviewer that MIC determinations have traditionally been performed in MHII medium. However, in this work, we decided to use LB mainly for consistency with previous works from the lab. Specifically, we have accumulated extensive knowledge of the physiological consequences of carrying pOXA-48 in different strains growing in LB^{16,17,19}, and we used LB in the work where we initially described plasmid-induced CS³⁰. Moreover, MIC data obtained in LB and MHII for different antibiotics show a strong, significant correlation (see below), which indicates that both media offer comparable results. We have incorporated this data as the new Supplementary Figure 14.

References

1. Spira, B., de Almeida Toledo, R., Maharjan, R. P. & Ferenci, T. The uncertain consequences of transferring bacterial strains between laboratories - rpoS instability as an example. *BMC Microbiol.* **11**, 248 (2011).
2. Mouton, J. W. *et al.* MIC-based dose adjustment: facts and fables. *J. Antimicrob. Chemother.* **73**, 564–568 (2018).
3. Charlton, C. L., Hindler, J. A., Turnidge, J. & Humphries, R. M. Precision of Vancomycin and Daptomycin MICs for Methicillin-Resistant *Staphylococcus aureus* and Effect of Subculture and Storage. *J. Clin. Microbiol.* **52**, 3898–3905 (2020).
4. Bhalodi, A. A., Oppermann, N., Campeau, S. A. & Humphries, R. M. Variability of Beta-Lactam Broth Microdilution for *Pseudomonas aeruginosa*. *Antimicrob. Agents Chemother.* **65**, 10.1128/aac.00640-21 (2021).
5. Mouton, J. W., Meletiadiis, J., Voss, A. & Turnidge, J. Variation of MIC measurements: the contribution of strain and laboratory variability to measurement precision. *J. Antimicrob. Chemother.* **73**, 2374–2379 (2018).
6. San Millan, A., Escudero, J. A., Gifford, D. R., Mazel, D. & MacLean, R. C. Multicopy plasmids potentiate the evolution of antibiotic resistance in bacteria. *Nat. Ecol. Evol.* **1**, 1–8 (2016).
7. Reguera, J. A., Baquero, F., Perez-Diaz, J. C. & Martinez, J. L. Synergistic effect of dosage and bacterial inoculum in TEM-1 mediated antibiotic resistance. *Eur. J. Clin. Microbiol. Infect. Dis.* **7**, 778–779 (1988).
8. Dimitriu, T., Matthews, A. C. & Buckling, A. Increased copy number couples the evolution of plasmid horizontal transmission and plasmid-encoded antibiotic resistance. *Proc. Natl. Acad. Sci.* **118**, e2107818118 (2021).
9. El Meouche, I., Siu, Y. & Dunlop, M. J. Stochastic expression of a multiple antibiotic resistance activator confers transient resistance in single cells. *Sci. Rep.* **6**, 19538 (2016).
10. Wang, X. *et al.* Heteroresistance at the Single-Cell Level: Adapting to Antibiotic Stress through a Population-Based Strategy and Growth-Controlled Interphenotypic Coordination. *mBio* **5**, 10.1128/mbio.00942-13 (2014).
11. Alonso-del Valle, A. *et al.* Antimicrobial resistance level and conjugation permissiveness shape plasmid distribution in clinical enterobacteria. *Proc. Natl. Acad. Sci.* **120**, e2314135120 (2023).
12. Shilling, P. J., Khananisho, D., Cumming, A. J., Söderström, B. & Daley, D. O. Signal amplification of araC pBAD using a standardized translation initiation region. *Synth. Biol.* **7**, ysac009 (2022).
13. ThermoFisher Scientific. pBAD/His A, B, and C pBAD/Myc-His A, B, and C. User Guide. https://www.thermofisher.com/document-connect/document-connect.html?url=https://assets.thermofisher.com/TFS-Assets%2FMSG%2Fmanuals%2Fpbad_man.pdf.
14. Guzman, L. M., Belin, D., Carson, M. J. & Beckwith, J. Tight regulation, modulation, and high-level expression by vectors containing the arabinose P(BAD) promoter. *J. Bacteriol.* **177**, 4121–4130 (1995).
15. Siegele, D. A. & Hu, J. C. Gene expression from plasmids containing the araBAD promoter at subsaturating inducer concentrations represents mixed populations. *Proc. Natl. Acad. Sci.* **94**, 8168–8172 (1997).
16. Alonso-del Valle, A. *et al.* Variability of plasmid fitness effects contributes to plasmid persistence in bacterial communities. *Nat. Commun.* **12**, (2021).

17. DelaFuente, J. *et al.* Within-patient evolution of plasmid-mediated antimicrobial resistance. *Nat. Ecol. Evol.* 2022 1–12 (2022) doi:10.1038/s41559-022-01908-7.
18. León-Sampedro, R. *et al.* Pervasive transmission of a carbapenem resistance plasmid in the gut microbiota of hospitalized patients. *Nat. Microbiol.* **6**, 606–616 (2021).
19. Fernández-Calvet, A. *et al.* The distribution of fitness effects of plasmid pOXA-48 in clinical enterobacteria: This article is part of the Microbial Evolution collection. *Microbiology* **169**, (2023).
20. Poirel, L., Bonnin, R. A. & Nordmann, P. Genetic features of the widespread plasmid coding for the carbapenemase OXA-48. *Antimicrob. Agents Chemother.* **56**, 559–562 (2012).
21. Hamprecht, A. *et al.* Pathogenicity of Clinical OXA-48 Isolates and Impact of the OXA-48 IncL Plasmid on Virulence and Bacterial Fitness. *Front. Microbiol.* **10**, (2019).
22. Carattoli, A., Seiffert, S. N., Schwendener, S., Perreten, V. & Endimiani, A. Differentiation of IncL and IncM Plasmids Associated with the Spread of Clinically Relevant Antimicrobial Resistance. *PLOS ONE* **10**, e0123063 (2015).
23. Potron, A., Poirel, L. & Nordmann, P. Derepressed Transfer Properties Leading to the Efficient Spread of the Plasmid Encoding Carbapenemase OXA-48. *Antimicrob. Agents Chemother.* **58**, 467–471 (2014).
24. Göttig, S., Gruber, T. M., Stecher, B., Wichelhaus, T. A. & Kempf, V. A. J. In Vivo Horizontal Gene Transfer of the Carbapenemase OXA-48 During a Nosocomial Outbreak. *Clin. Infect. Dis.* **60**, 1808–1815 (2015).
25. Ledda, A. *et al.* Hospital outbreak of carbapenem-resistant Enterobacterales associated with a blaOXA-48 plasmid carried mostly by Escherichia coli ST399. *Microb. Genomics* **8**, 000675 (2022).
26. Daikos, G. L. & Markogiannakis, A. Carbapenemase-producing *Klebsiella pneumoniae*: (when) might we still consider treating with carbapenems? *Clin. Microbiol. Infect.* **17**, 1135–1141 (2011).
27. Poirel, L., Potron, A. & Nordmann, P. OXA-48-like carbapenemases: the phantom menace. *J. Antimicrob. Chemother.* **67**, 1597–1606 (2012).
28. Walsh, T. R., Toleman, M. A., Poirel, L. & Nordmann, P. Metallo- β -Lactamases: the Quiet before the Storm? *Clin. Microbiol. Rev.* **18**, 306–325 (2005).
29. Walther-Rasmussen, J. & Høiby, N. Class A carbapenemases. *J. Antimicrob. Chemother.* **60**, 470–482 (2007).
30. Herencias, C. *et al.* Collateral sensitivity associated with antibiotic resistance plasmids. *eLife* **10**, 1–13 (2021).

REVIEWERS' COMMENTS

Reviewer #1 (Remarks to the Author):

The Authors have sufficiently answered all my questions and substantially improved the paper with the new experiments and analysis.

Reviewer #3 (Remarks to the Author):

The authors have made a thorough revision of the paper and addressed all my concerns. I congratulate them on a nice study!